# Itaconate and derivatives reduce interferon responses and inflammation in influenza A virus infection

**Aaqib Sohail**[1,2¤a‡], **Azeem A. Iqbal**[1,2‡], **Nishika Sahini**[1,2‡], **Fangfang Chen**[1,2‡], **Mohamed Tantawy**[1,2,3,4‡], **Syed F.H. Waqas**[1,2‡], **Moritz Winterhoff**[1,2‡], **Thomas Ebensen**[5], **Kristin Schultz**[6], **Robert Geffers**[7], **Klaus Schughart**[6,8,9], **Matthias Preusse**[1], **Mahmoud Shehata**[10,11], **Heike Bähre**[12], **Marina C. Pils**[13], **Carlos A. Guzman**[5], **Ahmed Mostafa**[10,11], **Stephan Pleschka**[10,14], **Christine Falk**[15], **Alessandro Michelucci**[16,17], **Frank Pessler**[1,2,18]*

**1** Biomarkers for Infectious Diseases, Helmholtz Centre for Infection Research, Braunschweig, Germany, **2** TWINCORE Centre for Experimental and Clinical Infection Research, Hannover, Germany, **3** Hormones Department, Medical Research and Clinical Studies Institute, National Research Center, Dokki, Giza, Egypt, **4** Stem Cells Lab, Center of Excellence for Advanced Sciences, National Research Center, Dokki, Giza, Egypt, **5** Vaccinology and Applied Microbiology, Helmholtz Centre for Infection Research, Braunschweig, Germany, **6** Infection Genetics, Helmholtz Centre for Infection Research, Braunschweig, Germany, **7** Genome Analytics, Helmholtz Centre for Infection Research, Braunschweig, Germany, **8** University of Veterinary Medicine Hannover, Hannover, Germany, **9** Department of Microbiology, Immunology and Biochemistry, University of Tennessee Health Science Center, Memphis, Tennessee, United States of America, **10** Institute for Medical Virology, Justus-Liebig-University, Giessen, Germany, **11** Center of Scientific Excellence for Influenza Viruses, National Research Centre, Giza, Egypt, **12** Research Core Unit Metabolomics, Hannover Medical School, Hannover, Germany, **13** Mouse Pathology Platform, Helmholtz Centre for Infection Research, Braunschweig, Germany, **14** German Center for Infection Research (DZIF) partner site Giessen, Germany, **15** Department of Transplantation Immunology, Hannover Medical School, Hannover, Germany, **16** Neuro-Immunology Group, Department of Cancer Research, Luxembourg Institute of Health (LIH), Luxembourg, **17** Luxembourg Centre for Systems Biomedicine, University of Luxembourg, Esch-sur-Alzette, Luxembourg, **18** Centre for Individualised Infection Medicine, Hannover, Germany

¤a Current address: Department of Medicine, Harvard Medical School, the Division of Allergy and Clinical Immunology, Brigham and Women's Hospital, Boston, Massachusetts

‡ AS, AI, and NS share first authorship on this work. FC, MT, FW and MW are joint second authors on this work (listed in alphabetical order).

* frank.pessler@helmholtz-hzi.de, frank.pesslerwincore.de

**Data Availability Statement:** The gene expression data are available through GEO (https://www.ncbi.nlm.nih.gov/geo) at accession numbers GSE162210, GSE162260, GSE162261 and

## Abstract

Excessive inflammation is a major cause of morbidity and mortality in many viral infections including influenza. Therefore, there is a need for therapeutic interventions that dampen and redirect inflammatory responses and, ideally, exert antiviral effects. Itaconate is an immunomodulatory metabolite which also reprograms cell metabolism and inflammatory responses when applied exogenously. We evaluated effects of endogenous itaconate and exogenous application of itaconate and its variants dimethyl- and 4-octyl-itaconate (DI, 4OI) on host responses to influenza A virus (IAV). Infection induced expression of ACOD1, the enzyme catalyzing itaconate synthesis, in monocytes and macrophages, which correlated with viral replication and was abrogated by DI and 4OI treatment. In IAV-infected mice, pulmonary inflammation and weight loss were greater in *Acod1*[-/-] than in wild-type mice, and DI treatment reduced pulmonary inflammation and mortality. The compounds reversed infection-triggered interferon responses and modulated inflammation in human cells supporting non-productive and productive infection, in

GSE164922. The data previously published in [31] are available at GSE66040.

**Funding:** We gratefully acknowledge support from Helmholtz Association of German Research Centres (award iMed/HZI and AMPro/HZI to FP), Federal Ministry for Science and Education (BMBF) award COVID-Protect (01KI20143C) to FP, German Academic Exchange Service (DAAD) predoctoral fellowships to AI and MT, Deutsche Forschungsgemeinschaft SFB 1021 (Project C01) to SP (Co-PI), German Center for Infection Research (DZIF) partner site Giessen (TTU 01.806) to SP (Co-PI), and the Alexander von Humboldt Foundation (Georg Forster Research Fellowship) to MS. The funders had no role in the study design, data collection and analysis, decision to publish, or preparation of the manuscript.

**Competing interests:** The authors have declared that no competing interests exist.

**Dedication**: This publication is dedicated to the memory of Lisa Pessler.

peripheral blood mononuclear cells, and in human lung tissue. All three itaconates reduced ROS levels and STAT1 phosphorylation, whereas AKT phosphorylation was reduced by 4OI and DI but increased by itaconate. Single-cell RNA sequencing identified monocytes as the main target of infection and the exclusive source of *ACOD1* mRNA in peripheral blood. DI treatment silenced IFN-responses predominantly in monocytes, but also in lymphocytes and natural killer cells. Ectopic synthesis of itaconate in A549 cells, which do not physiologically express *ACOD1*, reduced infection-driven inflammation, and DI reduced IAV- and IFNγ-induced *CXCL10* expression in murine macrophages independent of the presence of endogenous *ACOD1*. The compounds differed greatly in their effects on cellular gene homeostasis and released cytokines/chemokines, but all three markedly reduced release of the pro-inflammatory chemokines CXCL10 (IP-10) and CCL2 (MCP-1). Viral replication did not increase under treatment despite the dramatically repressed IFN responses. In fact, 4OI strongly inhibited viral transcription in peripheral blood mononuclear cells, and the compounds reduced viral titers (4OI>Ita>DI) in A549 cells whereas viral transcription was unaffected. Taken together, these results reveal itaconates as immunomodulatory and antiviral interventions for influenza virus infection.

## Author summary

Interferon responses are part of the primary host defenses against infections. However, excessive inflammation is often a major factor in severe disease or even death in respiratory infections such as influenza, as it can lead to acute respiratory distress syndrome and sepsis-like multiorgan involvement. We applied itaconate and chemically modified versions of it (which enter cells more efficiently and can be applied at lower doses) to influenza A virus-infected human cells and lung tissue and found that these compounds markedly repress interferon responses and some pro-inflammatory processes without increasing viral replication. In fact, 4-octyl itaconate greatly decreased viral RNA replication in peripheral blood, and itaconate and 4-octyl itaconate reduced production of infectious virus in a human lung cell line. By analyzing gene expression patterns of single mononuclear cells in peripheral blood, we found that the virus infects predominantly monocytes and that these cells are the only source of ACOD1, the enzyme that synthesizes itaconate in humans. In a mouse model of influenza A virus infection, dimethyl-itaconate prevented lung inflammation and improved survival. Thus, our results suggest that novel medications based on itaconate promise to be effective treatments for influenza because they reduce deleterious inflammation and potentially also limit viral spread in the patient.

## Introduction

As exemplified by influenza viruses [1] and pathogenic coronaviruses [2], overshooting host inflammation, oxidative stress and cell damage/death are major determinants of morbidity and mortality in severe viral infections of humans. Consequently, there is a great need for adjunct treatments that can be given in conjunction with direct-acting anti-virals, with the aim to modulate the host response in such a way that end-organ damage is reduced and clinical outcome is improved.

Itaconic acid initially attracted great interest in the field of biotechnology due to its highly reactive double bonds, which make it an efficient intermediate in the biosynthesis of industrial

polymers (reviewed in [3]). The discoveries that its synthesis is highly induced during macrophage activation [4] and that *immune response gene 1* (*Irg1*, since then renamed *aconitate decarboxylase*) encodes the enzyme cis-aconate decarboxylase (ACOD1 in humans, CAD in other organisms), which converts cis-aconitate to itaconate [5, 6], have triggered intense efforts to elucidate functions of the endogenous ACOD1/itaconate axis in inflammation and infection. The early recognition that itaconate possesses anti-inflammatory and immunomodulatory properties also suggested that itaconate and prodrugs derived from it may have potential as immunomodulatory and cytoprotective interventions for inflammation-driven diseases (reviewed in [7, 8]).

Most evidence of anti-inflammatory and immunomodulatory effects of itaconate has actually been obtained by treating cells or small animals with chemically modified variants that are presumably taken up more efficiently, but also may have different effects from unmodified itaconate. 4-Octyl-itaconate (4OI) contains an added 8-carbon chain and acts primarily through activation of the NRF2 pathway [9]. It has, for instance, shown cytoprotective and anti-inflammatory effects in a variety of models, including lipopolysaccharide (LPS)-induced macrophage activation and LPS-induced sepsis in mice [9], in a neuronal model of oxidative stress [10], and in peripheral blood mononuclear cells (PBMCs) from patients with systemic lupus erythematosus [11]. Dimethyl-itaconate (DI, in which both carboxyl groups are methylated) has also been shown to exert NRF2-independent anti-inflammatory effects by inhibiting IκBζ activity [12]. Like 4OI, DI has shown beneficial effects in a variety of mouse models of infection-associated inflammation such as LPS-induced mastitis [13] and endometritis [14], fungal keratitis [15], and others. Furthermore, DI possesses cytoprotective effects, which was, for instance, evidenced by reduced cell damage in mouse models of cardiac ischemia [16]. However, cytoprotective properties have also been demonstrated for unmodified itaconate, notably in a rat model of cerebral ischemic-reperfusion injury [17]. As opposed to 4-OI and DI, which do not inhibit succinate dehydrogenase (SDH) [18], itaconate likely exerts cytoprotective effects at least in part by reducing reactive oxygen species (ROS) generation via inhibiting SDH [16, 17, 19]. An additional anti-inflammatory effect of itaconate and 4OI was recently discovered in that both can directly inhibit activation of the NLRP3 inflammasome and subsequent release of IL-1β [20]. Consistent with the differences in their chemical structures, recent work has provided more and more evidence that itaconate, DI, and 4OI exert distinct effects on cell inflammation, whereby unmodified itaconate has the greatest propensity to also exert proinflammatory effects. For instance, Swain et al. showed that unmodified itaconate can actually enhance IFN-β secretion from murine BMDMs and rescue deficient IFN production in BMDMs from *Acod1*[-/-] (i.e. *Irg1*[-/-]) mice [18]. Following early evidence that *Acod1* transcription is strongly induced in severe influenza A virus (IAV) infection in mice [21], its impact on host susceptibility to viral infections has also been studied: protective effects of itaconate and 4OI on neurons exposed to RNA viruses have been reported [22–24], and 4OI has been shown to reduce infectivity of SARS-CoV-2, herpes simplex virus-1 and -2, vaccinia virus, and Zika virus, but influenza virus has not been examined [25]. On the other hand, in a recent study using a mouse model of IAV infection, deleting the *Acod1* locus did not affect weight loss or survival, whereas it clearly worsened the outcome of *Mycobacterium tuberculosis* infection in the same mouse strain [26]. This lack of effect on IAV infection was surprising because abnormally increased inflammation is usually associated with deleterious effects on the host during viral infections. For instance, it has been shown that overactive inflammatory responses are associated with decreased survival and increased weight loss in IAV-infected inbred mouse strains [27]. While the role of endogenous itaconate in protection from viral infections thus remains incompletely understood, the above results do suggest a strong potential of external application of itaconate or its derivatives to ameliorate inflammation and other deleterious

consequences of viral infections while at the same time potentially also inhibiting viral infectivity.

Focusing on IAV as one of the most important human respiratory viral pathogens [28], we therefore examined role and regulation of the endogenous ACOD1/itaconate axis in a mouse model of IAV infection and subsequently performed a detailed analysis of the modulatory effects of exogenous addition of itaconate and derivatives on inflammatory responses in IAV-infected human cell lines, explanted lung tissue, PBMCs (both at the bulk and single-cell levels), and a mouse model. We find that loss of ACOD1 and itaconate synthesis led to increased inflammation during infection and that exogenous itaconate, DI, and 4OI substantially reduced inflammation, most notably IFN responses, whereby some pro-inflammatory effects were seen with unmodified itaconate. In spite of the strong reduction in IFN responses, viral replication was not enhanced. In fact, both itaconate and 4OI exhibited direct anti-viral effects.

## Results

### *Acod1* expression and itaconate synthesis are induced in mouse lung during IAV-infection and limit pulmonary inflammation and disease severity

Compared with C57BL/6J, DBA/2J mice are more susceptible to IAV infection in that weight loss is more rapid, mortality is higher, and inflammation is more pronounced [29, 30]. We have shown previously that *Acod1* transcription in the first 48 h of infection is higher in IAV-infected lung from DBA/2J than from C57BL/6J mice [21]. A reanalysis of our RNAseq data from a longer time course [31] showed that this difference persisted at least through day 3 post infection (p.i.) and that *Acod1* expression nearly normalized by day 14 in the surviving strain (**Fig 1A**). This analysis also revealed a remarkable correlation with expression of *Tnfaip3* (encoding an anti-inflammatory deubiquitinase, also known as *A20*) and a weaker positive correlation with *Hmox1* and *2* (encoding inducible anti-oxidant enzymes), all of which are known to be co-regulated with *Acod1* [32, 33]. A separate experiment confirmed that itaconate concentrations were higher in lungs from DBA/2J than from C57BL/6J mice on day 3 p.i. and correlated well with the differences in HA and *Acod1* mRNA expression (**S1 Fig**). *Acod1* was the 4th most highly upregulated pulmonary mRNA 48 h p.i. in C57BL/6J mice, where it appeared in a cluster of transcripts pertaining to IFN responses and innate immunity (**Fig 1B**). These results underscored the close association of pulmonary *Acod1* expression and itaconate accumulation with systemic inflammation and oxidative stress in IAV-infected mice. We therefore tested whether targeted deletion of *Acod1* would affect inflammatory responses and host susceptibility to IAV infection. Itaconate was not detectable in lungs from IAV-infected *Acod1*$^{-/-}$ mice; it was highly elevated on days 7/8 p.i. in the *Acod1*$^{+/+}$ and, significantly less, in *Acod1*$^{+/-}$ mice, and returned to normal by day 14 (**Fig 1C**). Both weight loss and mortality were higher in *Acod1*$^{-/-}$ than *Acod1*$^{+/+}$ female C57BL/6N mice (**Fig 1D and 1E**), with greatest changes occurring around day 8, the time of maximal itaconate accumulation. Histologically assessed pulmonary inflammation was greatest in *Acod1*$^{-/-}$ mice and intermediate in *Acod1*$^{+/-}$ mice (**Fig 1F**). Thus, itaconate synthesis correlated with disease activity, and loss of *Acod1* was associated with more severe disease and higher pulmonary inflammation.

### IAV infection induces *ACOD1* expression in human PBMCs and macrophages

We then tested whether IAV infection increased *ACOD1* mRNA levels in human PBMCs and in macrophages differentiated from PBMCs into M1 or M2 phenotype with granulocyte-

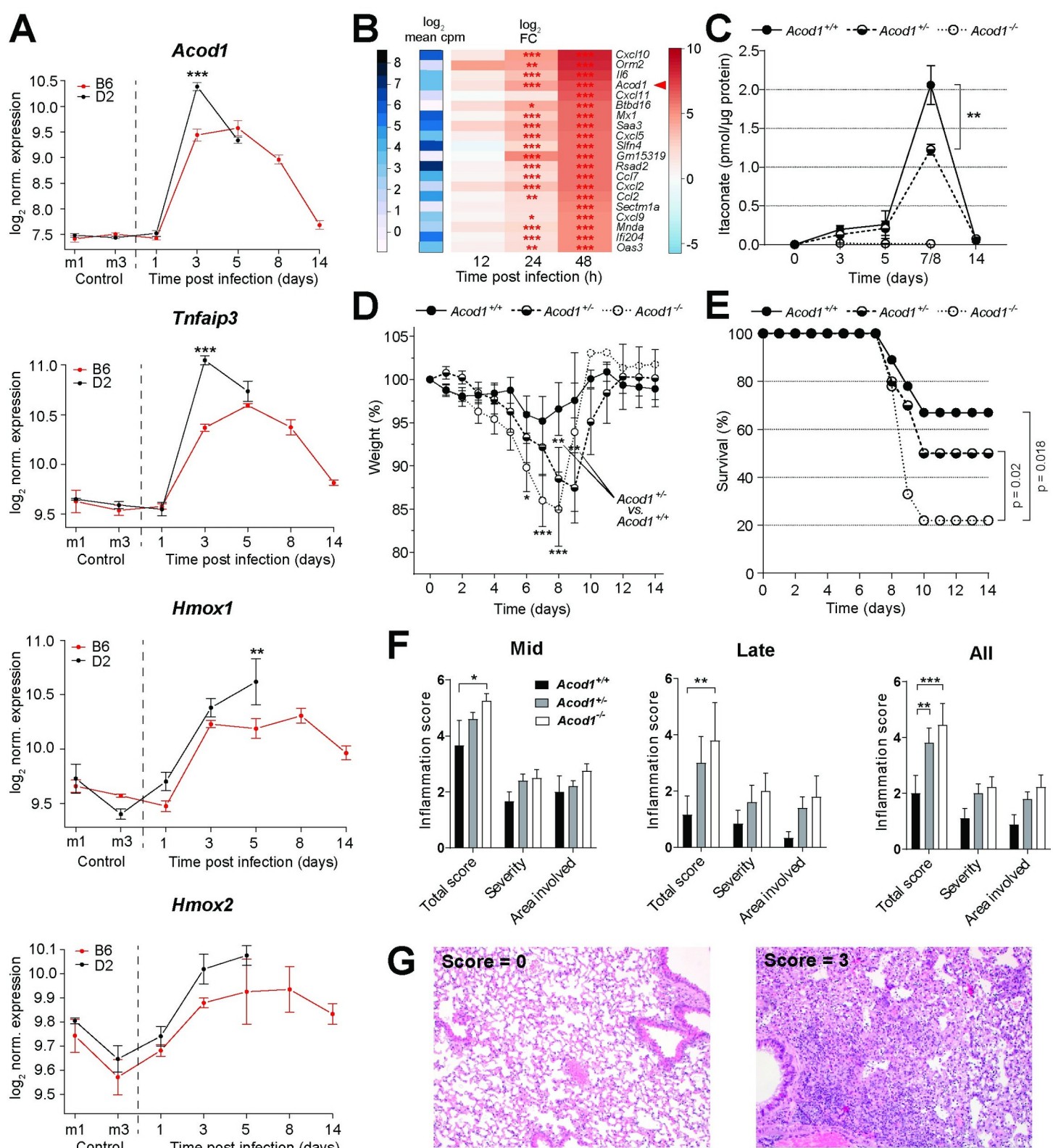

**Fig 1. Reciprocal interactions between *Acod1* expression and inflammation during influenza A virus infection in mice. A.** Higher pulmonary expression of *Acod1*, *Tnfaip3*, and *Hmox1* and *2* mRNA in DBA/2J (D2) than in C57BL/6J (B6) mice following IAV infection (2x10³ FFU PR8M; reanalysis of published RNAseq data [31])

(n = 3 each strain and time point). *p<0.05; **p<0.01; ***p<0.001 (pairwise *t*-test with Benjamini-Hochberg multiple testing correction). **B.** *Acod1* mRNA is among the most highly upregulated transcripts in mouse lung within 48 h of IAV infection (RNAseq, n = 3 per time point). **C.** Lower pulmonary levels of itaconate in IAV-infected *Acod1*[+/-] than in *Acod1*[+/+] mice. Itaconate was not detected in the *Acod1*[-/-] mice (n = 36 *Acod1*[+/+], 27 *Acod1*[+/-], 4 *Acod1*[-/-] mice). *p<0.05; **p<0.01; ***p<0.001 (unpaired t-test). **D.** Greater weight loss of female *Acod1*[-/-] C57BL/6N mice during infection with IAV. Mice had to be euthanized when weight loss exceeded 20%; n = 9 *Acod1*[+/+], 10 *Acod1*[+/-], 9 *Acod1*[-/-] mice. *p<0.05; **p<0.01; ***p<0.001 (unpaired t-test) **E.** Survival data of the same experiment as in C. Median survival after day 7 was significantly lower in *Acod1*[-/-] mice (p = 0.018, Mann-Whitney-U test). **F.** Semi-quantitative histopathological scoring of lungs on days 8/9 (mid), 11 and 14 (late) reveals highest inflammation in *Acod1*[-/-] mice; n = 9 *Acod1*[+/+], 10 *Acod1*[+/-], 9 *Acod1*[-/-] mice (all female). *p<0.05; **p<0.01; ***p<0.001 (two-way ANOVA, Tukey's multiple comparisons test) **G.** H&E stained lung sections showing inflammation severity scores of 0 (left, *Acod1*[+/+] mouse) and 3 (*Acod1*[-/-] mouse).

macrophage colony-stimulating factor (GM-CSF) and macrophage colony-stimulating factor (M-CSF), respectively. Phorbol myristate acetate (PMA)-differentiated THP-1 (dTHP-1) cells were analyzed for comparison. Viral transcription was highest in M2 cells (**Fig 2A**). The strongest *ACOD1* mRNA induction was seen in M2 cells and undifferentiated PBMC, and the weakest in M1 cells and dTHP-1 cells, even though the latter two cell types did support a significant degree of viral transcription. As in IAV-infected mouse lung (**Fig 1**), *TNFAIP3* and *ACOD1*

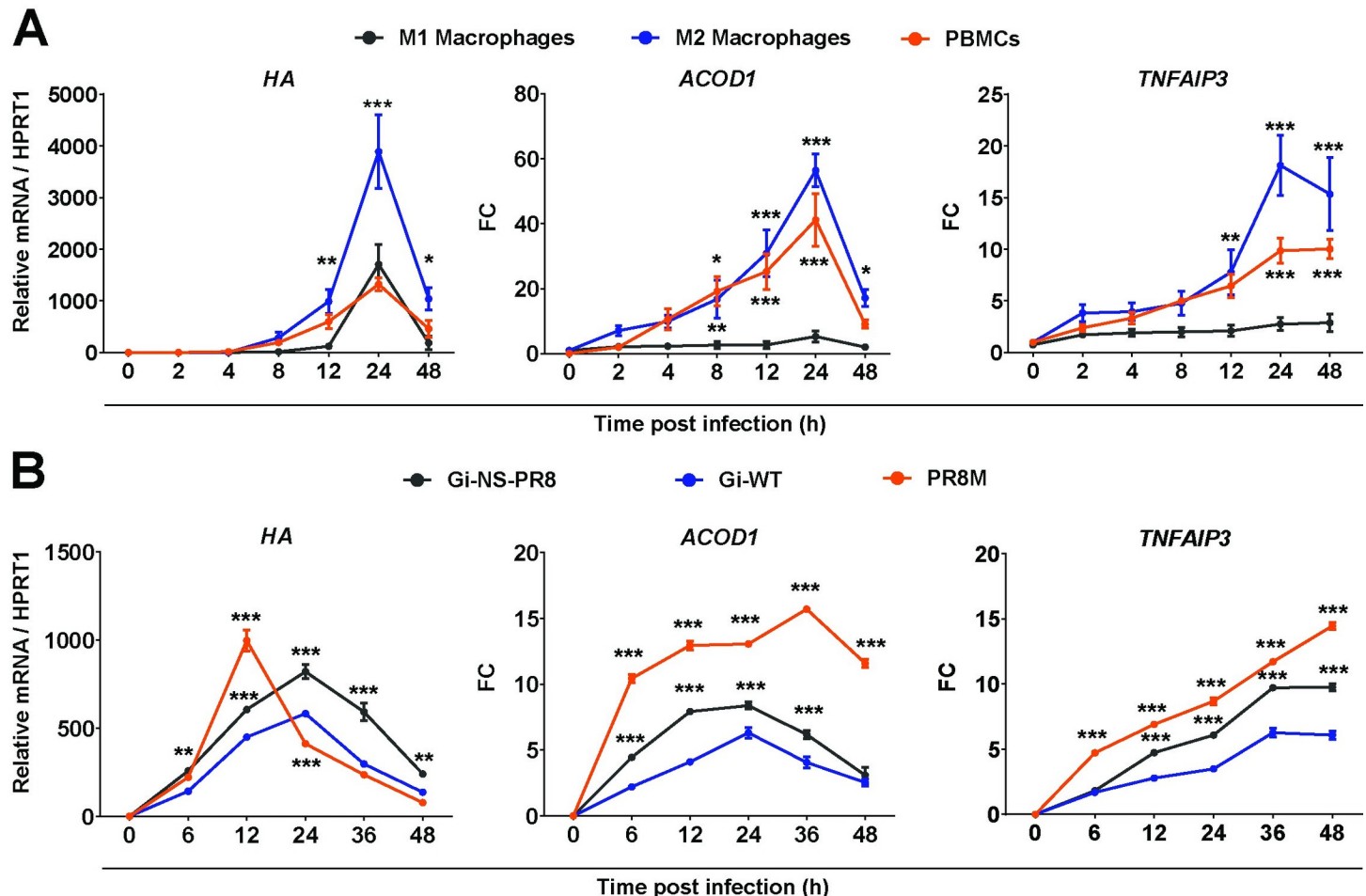

**Fig 2. Differences among human myeloid cells in *ACOD1* mRNA induction during IAV infection.** Cells were infected with the indicated IAV strains (MOI = 1) and gene expression determined at the time points post infection (p.i.) indicated on the x-axes. **A.** Induction of *ACOD1* and *TNFAIP3* mRNA (RT-qPCR) in M1 and M2 macrophages and PBMCs during infection with IAV (PR8M). **B.** Induction of *ACOD1* and *TNFAIP3* (RT-qPCR) in dTHP-1 cells correlates with virulence of three IAV strains (Gi-WT = H1N1(pdm2009) field isolate; G-NS-PR8 = H1N1(pdm2009) reassortant carrying NS segment of PR8; PR8M = H1N1(PR8M). Reference for fold change = uninfected cells at the same point. Mean ±SEM (n = 4). *HA* = IAV hemagglutinin. *p<0.05; **p<0.01; ***p<0.001 (two-way ANOVA, Tukey's multiple comparisons test, with reference to M1 macrophages in A, and to Gi-WT in B).

transcription correlated with each other. Infection of dTHP-1 cells with three IAV (H1N1) strains of differential replication efficiency [34] showed the strongest *ACOD1* and *TNFAIP3* induction by the two strains with the highest and the weakest by the strain with the lowest replication efficiency (**Fig 2B**). Thus, the extent of *ACOD1* induction after IAV infection differed significantly among the cell types tested, was strongest in M2 primary cells, was accompanied in all cases by an increase in *TNFAIP3* mRNA, and correlated with viral RNA replication.

## Itaconate and DI markedly reduce IFN responses in dTHP-1 cells without affecting viral RNA replication

After determining non-toxic concentrations (**S2 Fig**)**,** we then tested whether these compounds could modulate cellular inflammation due to IAV infection. For this purpose, we pre-incubated cells with the respective compound for 24 h, incubated the cells with virus-containing medium for 2 h, and then cultured the cells in fresh compound-containing medium for 12 h (**Fig 3A**). In infected dTHP-1 cells, treatments with itaconate or DI did not affect viral hemagglutinin (*HA)* mRNA expression, but both reduced induction of *CXCL10*, a key pro-inflammatory chemokine during influenza virus infection, with DI being several-fold more potent (**Figs 3B** and **3C and S3**). The 12 h time point was chosen because our previous work had identified it as the peak of IFN-I responses at the mRNA level in IAV-infected dTHP-1 cells [34]. We then assessed global effects of itaconate and DI on cellular gene mRNA expression. A comparison of differentially expressed genes (DEGs) suggested that the two compounds had both shared and unique effects on gene expression (**Fig 3D**). Both compounds had considerable effects on gene expression in uninfected cells (**Fig 3E and 3F**). As expected, IAV infection of untreated cells caused a major induction of IFN-regulated transcripts by 12 h p.i. (see mRNAs labeled in the volcano plot in **Fig 3G**). Strikingly, addition of both itaconate and DI led to global down-regulation of IFN-regulated genes. This was accompanied by upregulation of a substantial number of other transcripts, indicating that the compounds also modulated other cellular processes (**Fig 3H and 3I**). A principle component analysis (PCA) indicated that effects of the compounds on the cells were so pronounced at these concentrations that treated infected and treated uninfected cells clustered closely together, but far away from untreated infected or uninfected cells (**S4 Fig**). Indeed, a hierarchical clustering analysis revealed an across-the-board normalization of a clade of IFN- and inflammation-related transcripts by the compounds, but also clusters of DEGs that were upregulated by either or both compounds with respect to both uninfected and infected cells, further indicating effects upon cell homeostasis in general, but also suggesting that the two compounds had both common and unique effects on infected cells. For instance, both compounds reduced *CXCL10* expression, but only itaconate reduced *TGFB1* and *S100A8* expression (**S5 Fig**). mRNA and protein expression may be differentially affected by IAV infection due to virus-induced "host-cell shut-off". We therefore assessed release of inflammation-related polypeptides in supernatants from the same experiment (**Fig 3J**). This analysis confirmed the downregulation of pro-inflammatory factors (e.g., IP-10/CXCL10 and MCP-1/CCL2) by itaconate and DI, but also showed increased IL-8 (DI only) and minor increases of IL-1β (both itaconate and DI) in uninfected cells. Of note, both compounds also increased levels of the anti-inflammatory IL1 receptor antagonist (IL1RA) in supernatants from both infected and uninfected cells. A comprehensive view of the protein targets that were detected by this assay revealed that the two compounds effected substantially different changes in the target cells, as several factors were induced even in uninfected cells specifically by DI or itaconate (**S6A Fig**). At a later stage in the project, we compared effects of DI and 4OI on levels of the same targets in supernatants from IAV infected dTHP1 cells (**S6B Fig**). Compared to the first experiment, reducing effects of DI on

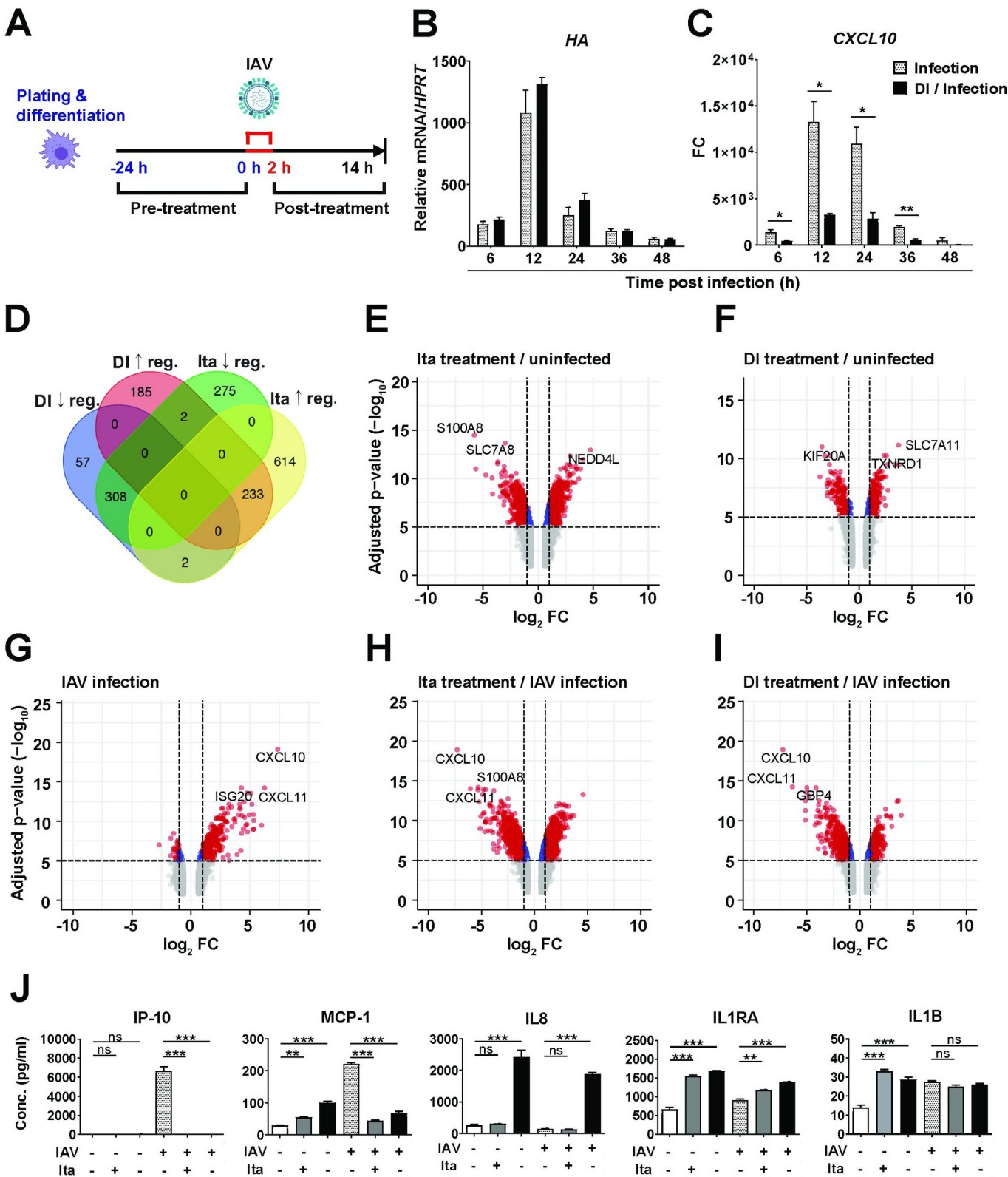

**Fig 3. Effects of itaconate and DI on cellular inflammation due to IAV infection of dTHP-1 cells. A.** Outline of experiment. dTHP-1 cells were treated with itaconate or DI overnight, or left untreated, and then infected with IAV (PR8M, MOI = 1). Dose-response curves of effects of a range of itaconate and DI

concentrations on *HA* and *CXCL10* mRNA expression are shown in **S3 Fig**. **B**-**C**. 48 h time course, DI = 0.25 mM (n = 3). mRNA expression (RT-qPCR) of *HA* (**B**) and *CXCL10* mRNA (**C**); reference for fold change = uninfected cells at the same time point. $^*p<0.05$; $^{**}p<0.01$; $^{***}p<0.001$ (unpaired t-test). **D-I**. Global transcriptomic changes (microarray analysis) 12 h p.i. due to itaconate (25 mM) and DI (1 mM) treatment of uninfected and IAV-infected dTHP-1 cells (n = 3). **D**. Venn diagram showing unique and common differentially expressed genes by itaconate and DI treatment of infected cells. **E-I**. Volcano plots showing differential mRNA expression in uninfected dTHP-1 cells under itaconate and DI treatment (**E-F**) and infected dTHP-1 cells with or without itaconate or DI treatment (**G-I**). The 2–3 most significantly differentially expressed mRNAs are identified by labels. **J**. Effects of itaconate (25 mM) and DI (1 mM) treatment on levels of inflammation-related polypeptides in dTHP-1 cell supernatants 12 h p.i. Multiplex cytokine/chemokine analysis of supernatants from the experiment shown in **D-I**. Mean ± SEM (n = 3). $^*p<0.05$; $^{**}p<0.01$; $^{***}p<0.001$ (One-way ANOVA, Tukey's multiple comparison test).

several targets were somewhat stronger, a reduction of IL-1β was also observed, and (in addition to IL-8 and IL-9) DI treatment increased levels of IL-10. These differences may have been due to differences in cell stock, reagent batch, or differentiation state of the cells. Importantly, 4OI exerted broader anti-inflammatory effects and did not increase levels of IL-8, 9, and 10. Thus, itaconate, DI, and 4OI apparently act on macrophages to release different recruitment programs that modify local inflammatory cell populations.

Analysis of enriched GO terms in the microarray underlying **Fig 3** (itaconate and DI treatment of dTHP1 cells) revealed that IFN-I signaling was strongly induced in the infected cells, which was prevented by treatment with both compounds (**S5B** and **S7** **Figs**). This analysis also revealed that both compounds dampened other pro-inflammatory processes, whereas itaconate also modulated processes relating to differentiation and membrane signaling. A KEGG pathway analysis revealed induction of classic pro-inflammatory pathways such as *TNF signaling*, *TLR receptor signaling*, and *chemokine signaling* by infection, all of which were depleted by itaconate and DI treatment (**S8 Fig**). Taken together, these results revealed strong anti-IFN and anti-inflammatory effects of itaconate and DI, but also suggested that they exert other important effects on cell metabolism and differentiation, some of which differ between the compounds.

## Anti-inflammatory effects of DI are independent of endogenous ACOD1 in primary murine macrophages

Considering that endogenous itaconate synthesis has been ascribed both pro- and anti-inflammatory effects [16, 18], we then tested whether the presence of endogenous ACOD1 would modulate the anti-IFN properties of DI. Bone marrow derived macrophages (BMDM) from WT and *Acod1*$^{-/-}$ mice were infected with IAV or stimulated with IFN-γ in the presence or absence of 0.25 mM DI (**Fig 4A–4C**). *CXCL10* mRNA expression at baseline was similar in WT and *Acod1*$^{-/-}$ cells. Both interventions vigorously induced *CXCL10* mRNA transcription, to a similar degree in WT and *Acod1*$^{-/-}$ cells. Likewise, treatment with DI reduced *CXCL10* expression to a similar degree in both genotypes, whereby the reducing effect was greater on IFN-γ stimulated than on IAV-infected cells. These results suggest that endogenous ACOD1 has no marked impact on *CXCL10* expression at baseline or after IAV infection/IFN-γ stimulation, and apparently does not play a role in the cellular anti-inflammatory response to DI treatment. In this experiment, we furthermore observed that the 6 h pre-incubation was sufficient to induce a significant down-regulation of *CXCL10*, but that the subsequent 12 h post-infection/stimulation treatment clearly amplified the effect. We therefore compared pre-incubation vs. post-infection treatment in the dTHP1 model (**Fig 4D and 4E**). In pre-incubation, 4OI greatly reduced *CXCL10* expression, whereas itaconate was ineffective. In post-infection treatment, 4OI was somewhat less effective, and itaconate reduced CXCL10 expression only at the higher dose. Taken together, these results underscore the differences between pure itaconate and 4OI and also reinforce the point made by Swain et al. [18] that the timeline of compound administration with regard to the stimulus needs to be considered when interpreting pharmacological effects of itaconates.

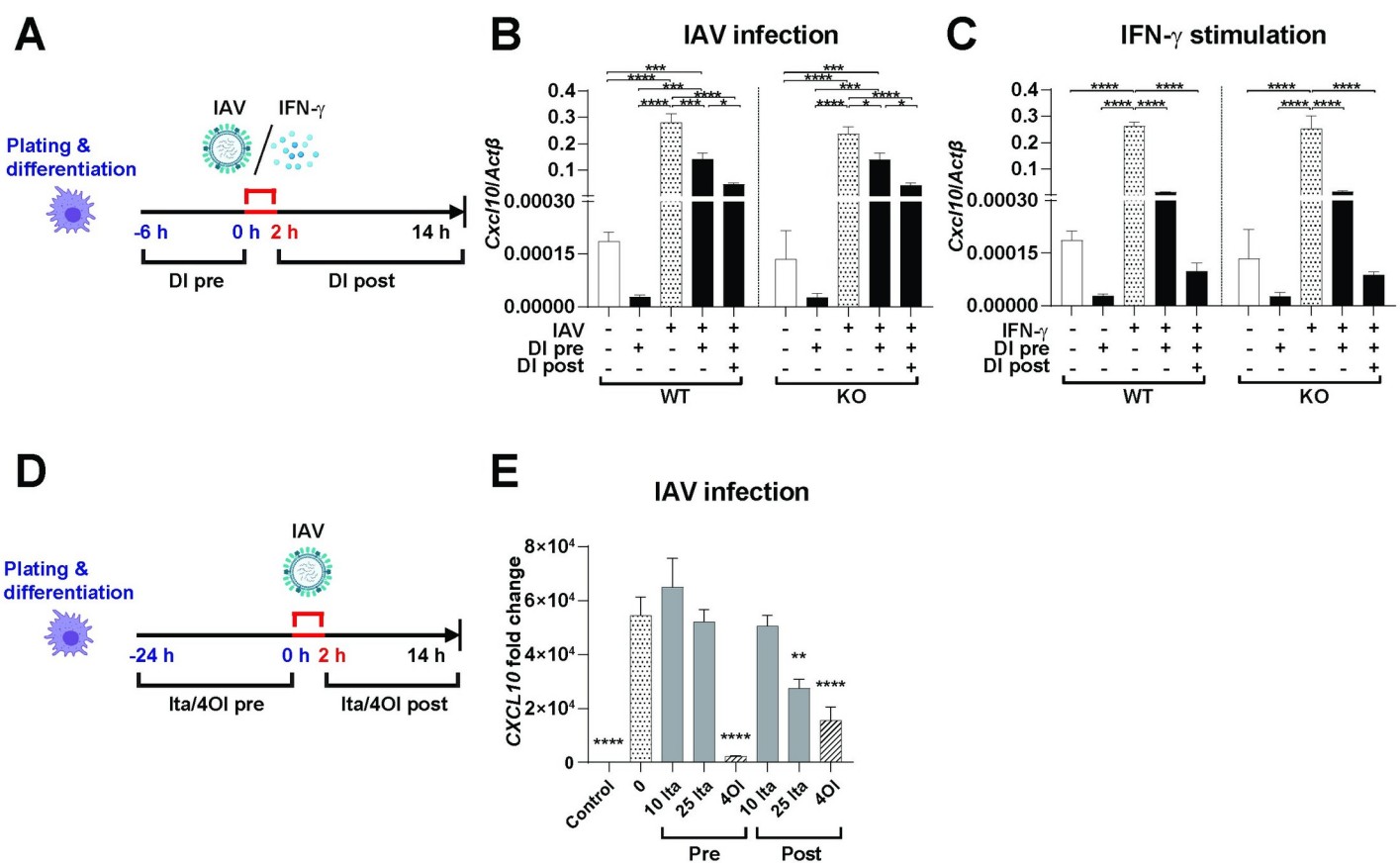

**Fig 4. Effects of endogenous *Acod1* (*Irg1*) gene and timing of compound administration on anti-inflammatory activity of itaconates. A-C. Effect of endogenous *ACOD1*.** BMDMs from WT or *Acod1*[-/-] C57BL/6N mice were preincubated with DI (0.25 mM) for 6 h, infected with IAV (PR8M, MOI = 1) or stimulated with IFN-γ (15 ng/ml) for 2 h, and then subjected to post-infection/stimulation treatment with DI (0.25 mM) for another 12 h as indicated. Uninfected/unstimulated cells were exposed to 6 h treatment only. Expression of *Cxcl10* mRNA was determined by RT-qPCR, using β-actin as internal control. **A.** Outline of experiment. **B.** *CXCL10* mRNA expression after IAV infection. **C.** *CXCL10* mRNA expression after IFN-γ stimulation. **D,E. Effect of timing of compound administration.** dTHP1 cells were infected with IAV (PR8M, MOI = 1) and subjected either to pre-incubation (24 h) or post-infection treatment (12 h) with itaconate (10, 25 mM) or 4OI (125 μM). Expression of *Cxcl10* mRNA was determined by RT-qPCR, using β-actin as internal control. Reference for fold change = uninfected, untreated 14 h. **D.** Outline of experiment. **E.** *CXCL10* mRNA expression. Mean ± SEM (n = 3). *p<0.05; **p<0.01; ***p<0.001; ****p<0.0001 (two-way ANOVA with Fisher's LSD test for multiple comparisons in **C,D**; one-way ANOVA with Dunnett's multiple comparison test in **C**).

## Anti-inflammatory effects of ectopically expressed itaconate

Considering that BMDMs from WT and *Acod1*[-/-] mice mounted comparable *CXCL10* responses (**Fig 4**), we then tested whether endogenously synthesized itaconate would affect inflammation in cells not naturally expressing *ACOD1*. For this purpose, we studied the respiratory epithelial cell line A549, which supports productive IAV infection and does not express *ACOD1*. As expected, infection of A549 cells did not result in induction of *ACOD1* mRNA, but transient transfection with an *ACOD1*-expressing plasmid led to significant *ACOD1* mRNA expression (**Fig 5A**), which was accompanied by appearance of substantial concentrations of itaconate. While there was no effect on viral *HA* mRNA expression, intracellular accumulation of itaconate led to a marked reduction in infection-associated *CXCL10* induction. Taken together with the results obtained in *Acod1*[-/-] BMDMs (**Fig 4**), these results suggest that immunomodulatory effects of itaconate and DI are independent of the ability of the cell type to express endogenous *ACOD1*. Furthermore, they show that similar effects on IFN-directed gene expression can be obtained with ectopic intracellular synthesis of itaconate and exogenously added itaconate and DI.

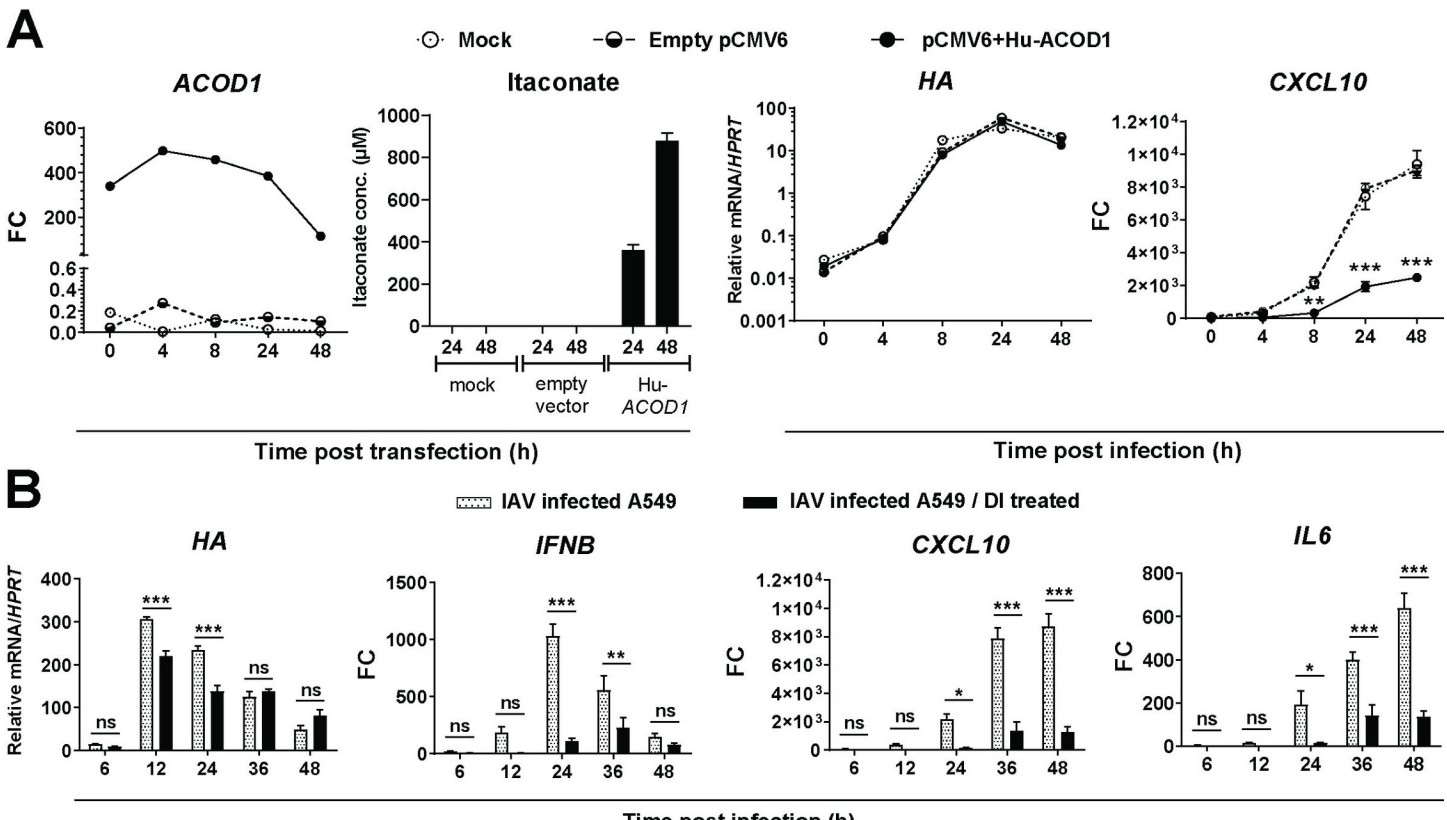

**Fig 5. Effect of overexpression of ACOD1 and exogenous treatment with DI on IAV-infected A549 cells. A.** A549 cells were transfected with pCMV6-Hu-ACOD1, pCMV6 empty vector, or mock transfected for 48 h and then infected with IAV (PR8M, MOI = 1) for 48 h (n = 3). Expression of the indicated mRNAs was measured by RT-qPCR, itaconate concentrations by LC-MS/MS. *p<0.05; **p<0.01; ***p<0.001 (2-way ANOVA, Tukey's multiple comparisons test) **B.** A549 cells were treated with 0.25 mM DI overnight, infected with IAV for 48 h (n = 3), and expression of the indicated mRNAs measured by RT-qPCR. Dose response curves of the effects of various concentrations of itaconate and DI on cell viability (MTT assay) are shown in **S2 Fig**. Reference for fold change = uninfected, untreated cells at the same time point. Mean ±SEM (n = 3). *p<0.05; **p<0.01; ***p<0.001 (2-way ANOVA, Sidak's multiple comparisons test).

## Anti-inflammatory effects of exogenous itaconate and DI on IAV-infected A549 respiratory epithelial cells

Since our treatment studies hat this far focused on myeloid cells, we tested whether exogenous application of itaconates would exert similar immunomodulatory effects on A549 cells, which support productive infection by influenza A viruses and resemble respiratory epithelial cells, i.e. important target cells of natural IAV infection in humans. A preliminary treatment experiment showed that (as opposed to THP-1 cells) DI was cytotoxic at 1.0 mM in A549 cells (also see **S2 Fig**), and the 0.5 mM concentration was therefore used. Treatment with DI resulted in a modest reduction of *HA* expression early in infection and a marked reduction of *CXCL10*, *IL6*, and *IFNB* expression (**Fig 5B**). Exogenous itaconate, too, reduced *CXCL10* mRNA, but less efficiently (**S9 Fig**).

Next, we assessed effects of itaconate and DI on cellular transcriptomes in IAV-infected A549 cells. PCA revealed the expected normalization of gene expression in IAV-infected cells by DI treatment, but also increasing effects of the compounds on overall cell homeostasis (**S10 Fig**). As in THP-1 cells, hierarchical clustering analysis revealed clear clustering of each of the four groups and the close relationship of the uninfected and DI-treated infected cells (**S11 Fig**). However, the anti-inflammatory effect of itaconate was weaker than that of DI, and at the

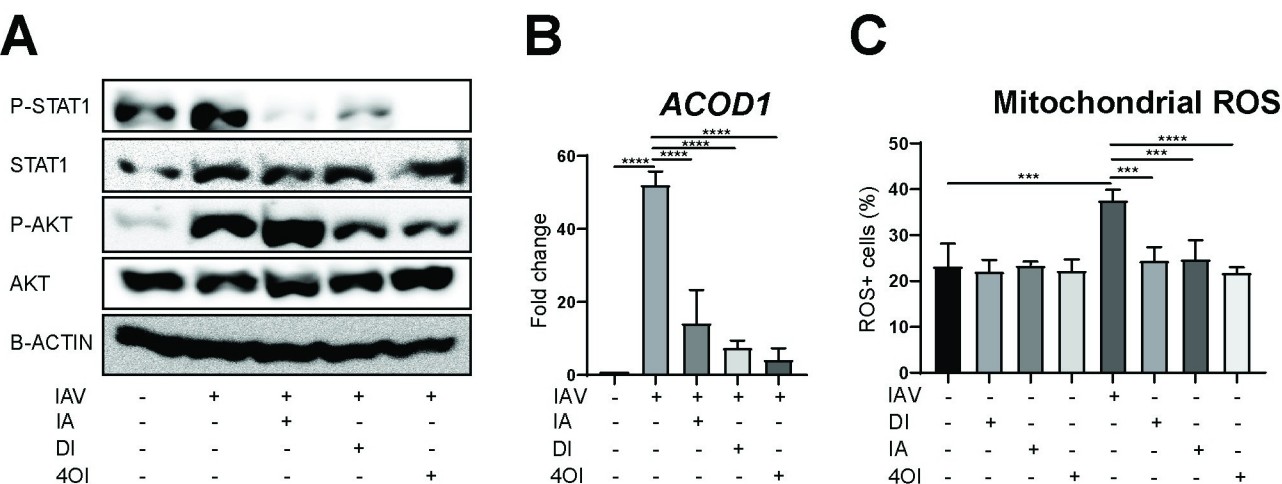

**Fig 6. Effects of itaconates on phosphorylation of STAT1 and AKT, *ACOD1* mRNA expression, and ROS levels.** Treatments (itaconate, 20 mM; DI 0.5 mM; 4OI 125 μM) were applied as indicated to dTHP1 cells infected with IAV (PR8M, MOI = 1), and analyses performed 12 h p.i. **A.** Itaconate, DI, and 4OI inhibit STAT1 phosphorylation, but only DI and 4OI inhibit AKT phosphorylation (immunoblot for the indicated targets, using β-actin as internal control). **B.** 4OI exerts the strongest *ACOD1* mRNA reduction (RT-qPCR, n = 3). Reference for fold change = uninfected cells 12 h. **C.** The three itaconates reduce IAV-induced mitochondrial ROS levels to a similar degree (flow cytometry, n = 3). Mean ±SEM *p<0.05; **p<0.01; ***p<0.001 (1-way ANOVA followed by Tukey's multiple comparison test).

higher (40 mM) concentration, the itaconate signature became so dominant that the itaconate-treated infected group now formed its own clade (**S12 Fig**). GO terms which were enriched in IAV infection related to IFN-I and–II responses as well as other antiviral and pro-inflammatory terms, which could all be reduced to near baseline by treatment with DI or itaconate (**S13A Fig**). Visualization of differentially expressed IFN-related mRNA also showed that, in spite of its overall lower effect on pro-inflammatory signaling than DI, itaconate treatment at the lower concentration (20 mM) did have a considerable impact on IFN-I expression (**S13B Fig**). In addition, itaconate uniquely inhibited *NFKB signaling* and *fatty acid metabolism*. KEGG-pathway analysis additionally revealed depletion of other pro-inflammatory pathways, including *TNF signaling* and *necroptosis*, by the treatments (**S14 Fig**).

## Itaconate, DI and 4OI inhibit STAT1 phosphorylation, but only DI and 4OI reduce AKT phosphorylation

In order to elucidate mechanisms underlying the immunomodulatory effects and differences among itaconate, DI, and 4OI, we assessed effects of the compounds on expression of *ACOD1* mRNA, phosphorylation of STAT1 (as a major mediator of classic IFN-I signaling) and AKT (as a well-known part of the cellular inflammatory response to IAV infection [35] and a mediator of non-canonical IFN-I signaling [36]) in dTHP1 cells. Infection led to a marked increase in P-STAT1 levels, which decreased markedly due to treatment with all three compounds, with 4OI showing the strongest effects (**Fig 6A**). This correlated with the ability of the compounds to suppress *ACOD1* mRNA expression, in that suppression was greatest by 4OI (**Fig 6B**). Likewise, P-AKT levels increased during infection and decreased significantly upon treatment with DI and 4OI; however, itaconate treatment actually led to a further increase in P-AKT (**Fig 6A**).

## Comparable anti-oxidative effects of itaconate, DI, and 4OI

Oxidative stress is a major pathogenic factor in influenza virus infection [37]. We therefore compared the effects of the three compounds on mitochondrial ROS levels. Infection led to a

marked increase in ROS+ dTHP1 cells, which was prevented by treatment with all three compounds (**Fig 6C**). Taken together, the results in **Fig 6** suggest (i) that the anti-IFN-I effect of unmodified itaconate can be explained by inhibition of IFNAR signaling, whereas inhibition of AKT signaling may further contribute to IFN-I reduction by DI and 4-OI, (ii) that some of the differences between itaconate and DI on cytokine/chemokine populations may be due to their opposite effects on P-AKT signaling, and (iii) that the three itaconate variants exert by and large similar anti-oxidative effects in IAV infection.

## Anti-inflammatory effect of itaconate and DI in a human lung tissue explant model of IAV infection

To test whether the *ACOD1*/itaconate axis and the anti-inflammatory effects of itaconate and DI are functional in the end-organ of IAV infection in humans, we then used a human lung tissue culture model. Indeed, infection with IAV led to brisk transcription of viral *HA* and host *CXCL10* mRNA in a 72 h time course (**S15 Fig**), whereas *ACOD1* mRNA levels increased only in a subset of tissue pieces. Adding itaconate and DI reduced expression of *CXCL10* mRNA in tissue and IP10 protein in culture supernatant, and itaconate additionally reduced ISG15 mRNA in tissue, whereas viral titers in the supernatant were unaffected (**Fig 7**). Re-analysis of a published dataset from a model using a different IAV strain (H3N2) and histologically normal lung tissue [38] revealed a pronounced induction of *ACOD1* and *TNFAIP3* expression, but (as opposed to *in vivo* infection of mouse lung) not induction of *HMOX1* or *2* (**S16 Fig**).

## Anti-inflammatory effects of itaconate, DI, and 4OI on IAV-infected PBMCs

PBMCs play important roles in influenza infection as they contain cells of innate and adaptive immunity, and because circulating monocytes (which can sustain a nonproductive influenza

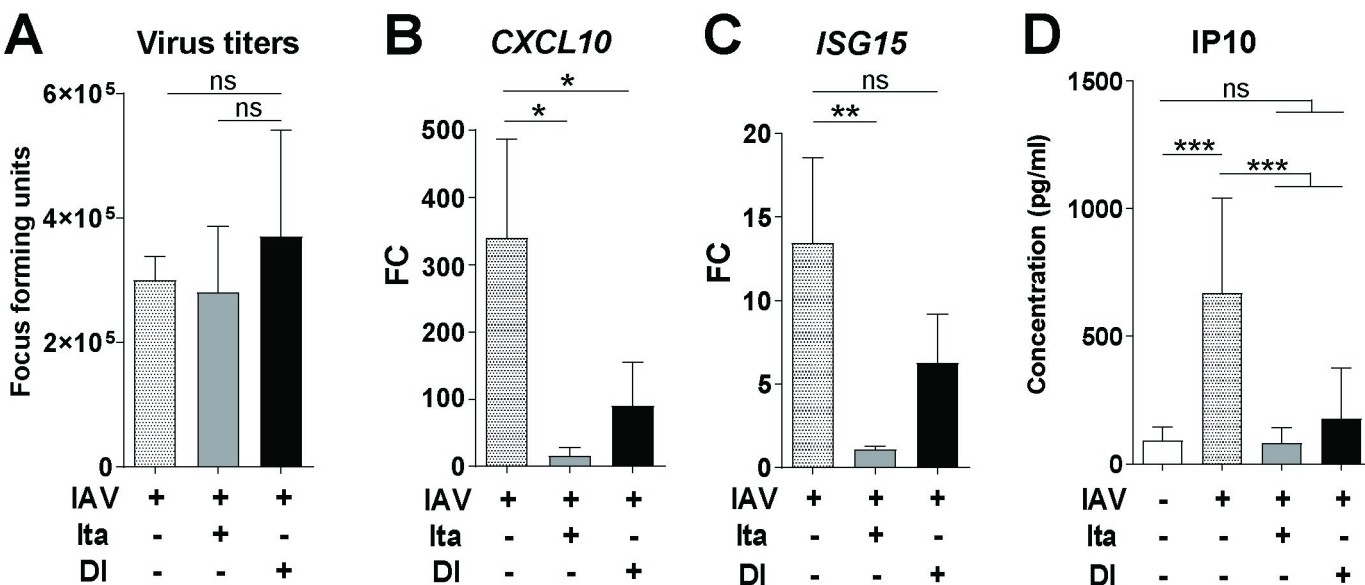

**Fig 7. Anti-inflammatory effect of itaconate and DI on IAV-infected human lung tissue explants.** Human primary lung tissue from patients with emphysema and pulmonary arterial hypertension was incubated overnight with itaconate (25 mM), DI (1 mM), or medium only and was then infected with IAV (PR8M, 2x10^5 FFU/ml) for 24 h. **A**. Viral titers in supernatant (focus forming assay). **B-D**. *CXCL10* mRNA (**B**), and *ISG15* (**C**) mRNA in tissue (RT-qPCR); IP10 concentration in supernatant (EIA) (**D**). Mean ± SEM (n = 6 donors, 3 tissue pieces per replicate). Reference for fold change = uninfected 24 h. *p<0.05; **p<0.01; ***p<0.001 (Mann-Whitney U test).

virus infection) are considered an important source of cytokines that support systemic spread of inflammation [39]. We therefore investigated effects of the compounds on IAV infected PBMCs both at the bulk and single cell levels. Analysis with targeted PCR- and immuno-assays demonstrated vigorous transcription of viral RNA, induction of *ACOD1* mRNA, and a brisk pro-inflammatory response both at the mRNA and protein level (**Fig 8A and 8B**). The induction of *ACOD1* mRNA expression in peripheral blood in IAV infection was also corroborated by reanalysis of a published dataset from whole blood from patients with moderate and severe influenza (**S17 Fig**). The effects of itaconate in our experiment were mixed in that there was no significant change in *ACOD1*, *IFNB1*, and *TNF* mRNA expression in cells or of IL1B protein in supernatants. In fact, itaconate treatment increased expression of *ACOD1* in uninfected cells (**Fig 8**). In contrast, DI and 4OI led to a marked reduction of IFN and pro-inflammatory cytokine expression at the mRNA and protein level, as well as of *ACOD1* mRNA. As exemplified by the effect of DI and 4OI on *TNF* and *ACOD1* expression, the treatments reduced expression of some mRNA targets even in uninfected cells, suggesting that they also reduced some baseline inflammation. Of note, 4OI in addition led to a pronounced (95%) reduction in *HA* transcription (p = <0.0001), indicating inhibition of viral RNA replication. PCA of bulk PBMC transcriptomes revealed strong effects of the compounds on cell transcriptomes in general, which led to one outlying DI-treated sample and two itaconate-treated samples, whereas all four 4OI-treated samples formed a clearly discernible group (**S18 Fig**). The most significant transcriptome changes were driven by 4OI, which had major effects on many genes that were not regulated by IAV infection or itaconate or DI treatments (**S19 Fig**). Nonetheless, induction of *HMOX1* by DI in IAV-infected samples was evident. Functional enrichment analysis of GO terms based on the transcriptomes revealed that infection induced the expected antiviral responses, including *response to IFN-I*, as well as several other inflammation-related terms (**Fig 8C**). At least one of the three compounds prevented activation of each of the infection-associated terms. However, consistent with its lack of effect on *IFNB* expression (**Fig 8**), itaconate did not affect *response to IFN-I* or *influenza A*. Several terms that were depleted by the compounds in infected PBMCs were not enriched by infection alone, suggesting that the compounds diminished also a baseline activation of the PBMCs. Interestingly, 4OI treatment strongly enriched terms relating to chromatin conformation, which turned out to be driven by upregulation of histone deacetylase expression. A KEGG pathway analysis essentially confirmed the findings of the GO term analysis (**S20 Fig**), including that 4OI enriched pathways relating to chromatin structure such as *Alcoholism*. Taken together, these results consolidated our above findings that itaconate and derivatives can modulate inflammation due to IAV infection, but also suggested that DI and 4OI may have more robust anti-IFN and anti-inflammatory effects than itaconate.

## Single-cell RNAseq identifies monocytes as the main PBMC host cell of IAV infection, sole source of *ACOD1* expression, and predominant target of immunosuppressive effects by DI

In order to characterize immunomodulation by an itaconate in PBMC subtypes, but excluding changes arising from anti-viral effects, we subsequently focused on DI to decipher its effects on gene expression at the single-cell level. Thirteen cell types could be identified in the PBMC scRNAseq data set, but we focused the analyses on the overarching five cell types: CD4+ and CD8+ T cells, B cells, NK cells, and monocytes (**Fig 9A and 9B**). Monocytes constituted the least numerous cell type, and DI treatment of control and infected cells led to a further reduction (**S21 Fig**). It was not possible to discern whether this was due to decreased survival or a technical artefact. Monocytes were essentially the only cell type expressing viral RNA, as only

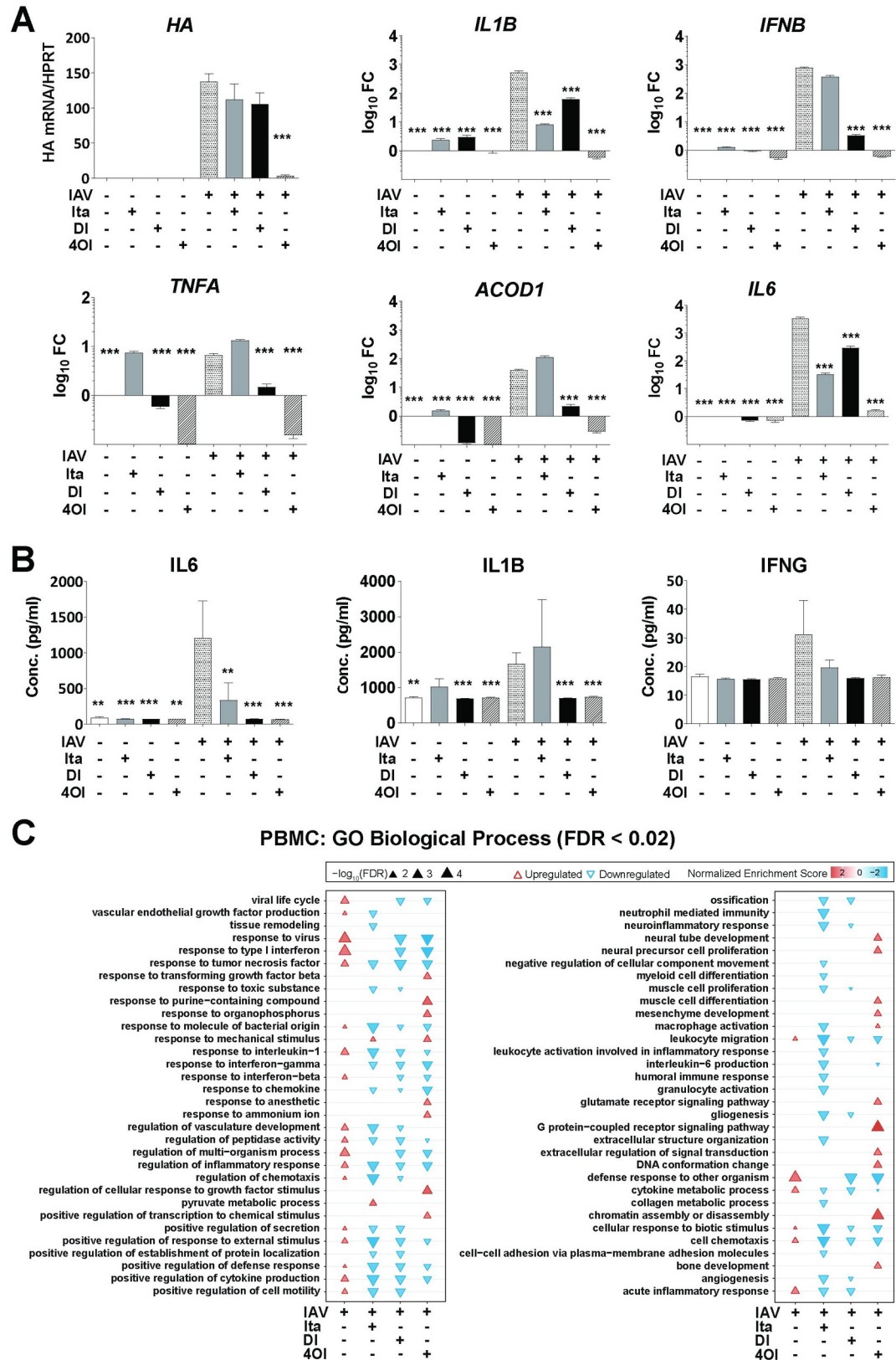

**Fig 8. Anti-inflammatory effects of itaconate, DI, and 4OI on IAV-infected human PBMCs.** PBMCs were isolated from freshly donated human blood and then infected with IAV (PR8M, MOI = 1) for 12 h in the presence or absence of itaconate

(10 mM), DI (0.5 mM), or 4OI (25 μM) without preincubation. **A.** Expression of the indicated mRNAs in PBMCs (RT-qPCR). Reference for fold change = uninfected, untreated 12 h. **B.** Concentrations of the indicated proteins in PBMCs supernatants (ELISA). Mean ±SEM (7 donors, 3 replicates per donor). $^*$p<0.05; $^{***}$p<0.01; $^{***}$p<0.001 (1-way ANOVA). **C.** GO enrichment analysis of effects of itaconate, DI, and 4OI on IAV-infected PBMCs. Microarray analysis of PBMC RNA from four of the donors featured in **A-B**. RNA was pooled from the three replicates of each donor, resulting in 4 samples per group. GO term analysis was performed on DEGs (p<0.05, FC>|1.5|) and terms with an FDR <0.02 in at least one condition are shown.

negligible amounts of viral transcripts were detected in lymphocytes and NK cells. DI treatment appeared to result in a slight increase in viral RNA expression in monocytes (**Fig 9C**). IAV infection upregulated expression of *ACOD1* exclusively, and *CXCL10* predominantly, in monocytes, whereas *TNFAIP3*, *IFIT1*, and *ISG15* were upregulated also in the NK, T and B cell compartments (**Fig 9D**). DI treatment markedly reduced expression of *ACOD1*, *IFIT1*, and *CXCL10*, but–interestingly–it reduced *TNFAIP3* and *ISG15* (albeit weakly) expression only in monocytes. Consistent with the observation that they constituted the main PBMC host cell, IAV infection triggered the most vigorous transcriptomic host response in monocytes, which was driven by the expected IFN responses (**Fig 9E**). However, discernable differential expression (mostly of IFN-related RNAs) was also observed in the other cell types. CD8+ T cells are shown as a representative example in **Fig 9F**, the other cell types in **S22 Fig**. Monocytes were the only cell type that mounted a transcriptomic response to DI treatment of uninfected cells. In infection, treatment with DI led to a global downregulation of IFN-related transcripts in monocytes and also downregulation of a subpopulation of IFN-related transcripts in lymphocytes and NK cells. When considering all DEGs irrespective of magnitude of differential expression, DI exerted comparable effects on monocytes and T cells, and only somewhat less on B cells. However, when considering only transcripts with FC >|2|, a preferential effect on monocytes became apparent (**Fig 9G and 9H**).

GO term enrichment analysis of the monocyte scRNAseq data confirmed the major enrichment of IFN-related processes by infection and their depletion by DI treatment, which had been seen in all models in this study. In addition, infection broadly dampened RNA metabolism, ribosome assembly, and protein synthesis and export, which was consistent with the cytostatic effects of type I IFN. DI treatment reversed all these effects and also stimulated some in uninfected cells (**Fig 10A**). A KEGG enrichment analysis of these monocyte data confirmed the depletion of IFN-related pathways (*influenza*, *measles*, *cytosolic DNA sensing pathway*) by DI treatment, but also of other pro-inflammatory pathways (*TLR signaling*, *cytokine-cytokine receptor interaction*, *NF-kB signaling*) in infected and uninfected cells (**S23 Fig**). Of note, DI treatment apparently stimulated ribosome function in both infected and control monocytes.

The Venn diagrams in **Fig 10B** illustrate that monocytes are the cell type in which the most biological processes were activated by infection and depleted by DI treatment, but reduction of antiviral and pro-inflammatory responses was a common theme in the other cell types as well (**Fig 10C**). Indeed, GO term analysis identified four major IFN-related processes as a central functional network that was upregulated by infection and downregulated by DI treatment during infection in all five cell types (**Fig 10B**). A marked stimulation of protein synthesis and export by DI was also apparent in the uninfected lymphocytes and NK cells. Finally, in order to identify a common transcriptomic network we performed GO and KEGG pathway enrichment analyses of the 81 transcripts that were commonly differentially expressed in IAV infection and regulated in the opposite direction by DI treatment in all five cell types (center of the Venn diagram in **Fig 9G**). While the GO analysis by and large confirmed the findings shown in **Fig 10B,** the KEGG pathway analysis revealed a major effect of DI treatment on *allograft rejection*, *graft-vs.-host disease*, and *antigen processing and presentation*, suggesting a broader interference with immune cell activation by DI (**S24 Fig**).

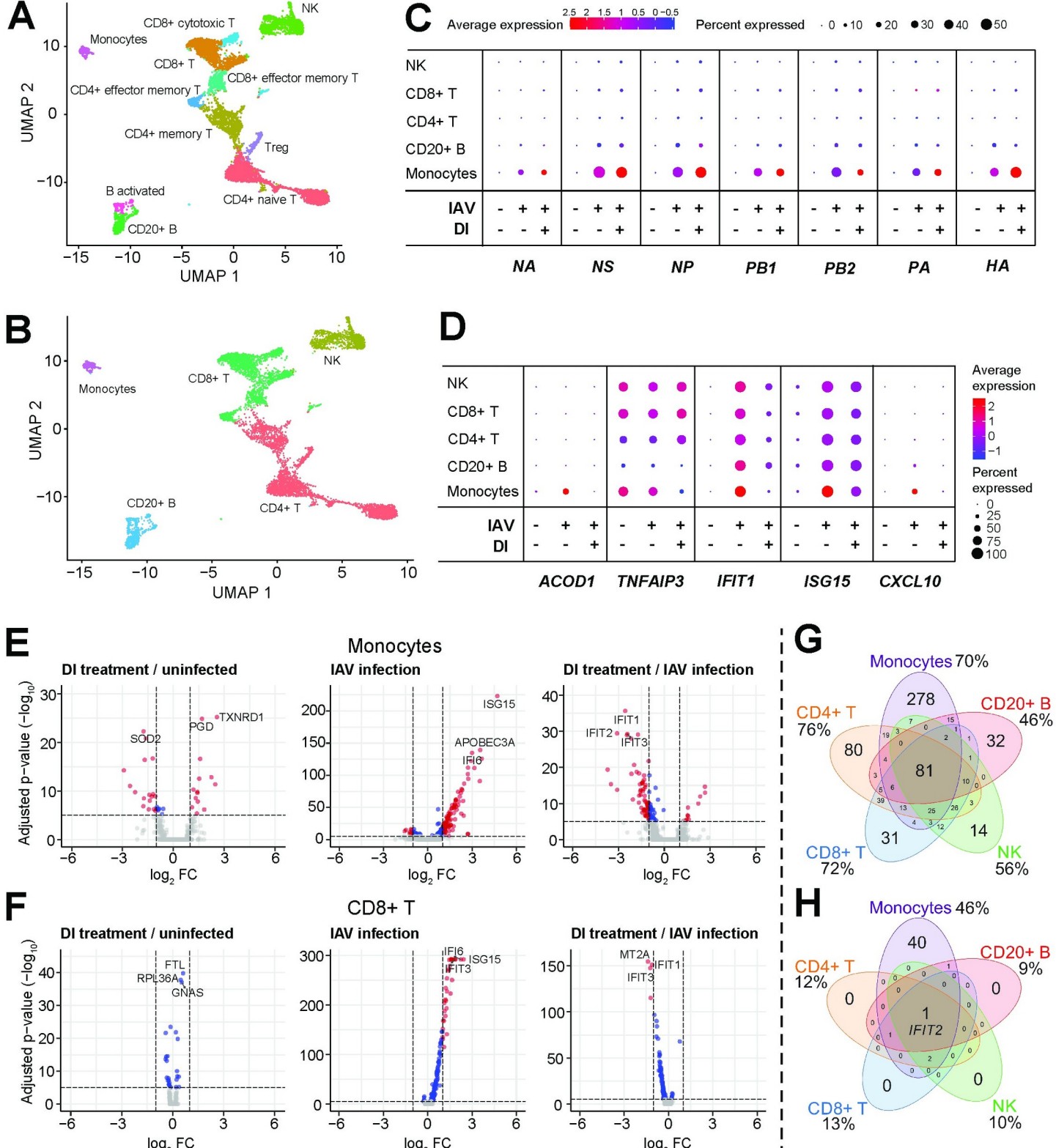

**Fig 9. Single-cell transcriptomic responses of PBMCs to IAV infection and DI treatment.** PBMCs were isolated from freshly donated human blood (one donor), and then infected with IAV (PR8M, MOI = 1) for 12 h with or without DI (0.5 mM) in the medium. **A,B**. UMAP plots identifying 13 cell types (**A**) that make up the five major cell types (**B**) studied. The RNA cell markers used are listed in **Table 4**. **C,D.** Differential mRNA expression in uninfected, infected and infected/DI-treated PBMCs. **E.**

Differential mRNA expression in monocytes in the indicated paired comparisons. Major transcriptional reprogramming due to IAV infection is evident, which is largely prevented by treatment with DI. **F.** Differential mRNA expression in CD8+ T cells in the indicated paired comparisons. Reprogramming in the presence of IAV is much less, but a DI treatment effect is evident. **G,H.** Venn diagrams based on RNAs differentially regulated (FDR <0.05) by both viral infection and DI treatment in each of the 5 major cell types, either irrespective of FC (**G**) or FC >|2| (**H**).

### Itaconate and 4OI reduce production of progeny virions in productive IAV infection

Considering the strong reduction of HA mRNA by 4OI in PBMCs we tested whether it would have anti-viral effects in productive infection in A549 cells as well. Surprisingly, it did not affect HA mRNA levels (**Fig 11A**). However, it greatly reduced viral titers in cell culture supernatants 24 h p.i., and a marked reduction also resulted from itaconate, but only a small reduction (approx. 50%) by DI (**Fig 11B and 11C**). Thus, of the 3 examined itaconates, 4OI clearly has the strongest anti-viral effects against IAV, the mechanisms of which differ depending on the host cell type.

### DI treatment prevents pulmonary inflammation and improves survival in the mouse model of IAV infection

Based on the above results that DI had strong anti-inflammatory but only minor antiviral properties, we then used the mouse model to test whether DI would reduce IAV-induced inflammation and whether any anti-inflammatory effect would lead to an increase in viral replication *in vivo*. Survival of DI-treated mice was significantly better (**Fig 11D**) but, surprisingly, weight loss did not differ between treated and untreated mice (**Fig 11E**). DI treatment essentially prevented pulmonary inflammation (**Figs 11F and S25**) but did not affect viral titers. While the lack of effect on weight loss remains to be explained (the most plausible hypothesis being that it is driven by ongoing viral replication), these results did demonstrate that DI exerted similar effects *in vivo* as in cells and lung tissue explants and that it primarily targeted host inflammation and not viral replication.

## Discussion

In this first dedicated study on the impact of itaconate and derivatives on host responses to IAV infection, we found that (i) the endogenous ACOD1/itaconate axis limits pulmonary inflammation due to the infection in a murine model, (ii) ectopically expressed ACOD1 possesses anti-IFN activity in an epithelial cell line that normally does not express this enzyme, and (iii) that exogenously applied itaconate and/or derivatives restrict IFN-responses and modulate classic pro-inflammatory cytokines across all models tested, whereas antiviral effects were most strongly associated with 4OI. Table 1 provides a comparison of the three compounds with respect to their major activities and effects.

### Anti-inflammatory effects of itaconate and derivatives

A major suppression of IFN and selected pro-inflammatory cytokine responses was the recurrent theme in all models used. This is fully consistent with the early reports that DI has strong anti-inflammatory properties [16], which was subsequently corroborated in diverse models of inflammation and infection, also using 4OI (e.g., [13, 15, 40]). We used itaconate and DI in parallel in several models, which enabled comparing their relative effectiveness. Overall, results with DI were most consistent, and in spite of its higher toxicity on A549 cells than on dTHP-1 cells, the lower dose that had to be used on A549 cells still possessed strong anti-inflammatory effects. Itaconate effects were less consistent and much higher doses had to be used to achieve

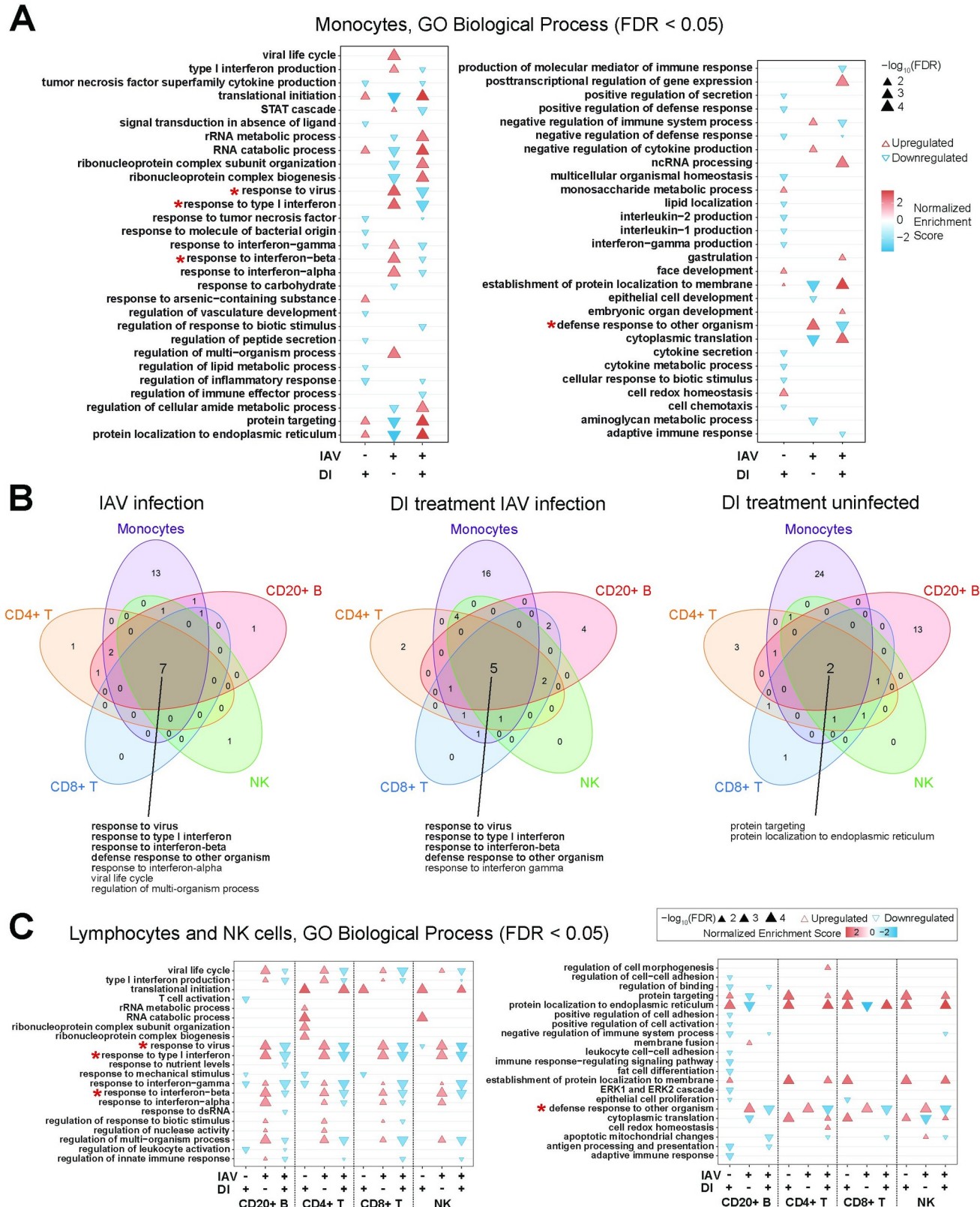

**Fig 10. GO enrichment analysis at the single cell level of DI effects on uninfected and IAV-infected PBMCs.** Analysis based on the scRNAseq data shown in Fig 9. Go terms that are enriched/depleted in all cell types by infection and DI treatment are marked with a red asterix in A and C. **A.** GO

enrichment analysis of monocytes. GO terms are listed in reverse alphabetical order from top to bottom, with upright (red) triangles indicating enrichment (upregulation) and downward pointing (blue) triangles depletion (downregulation). **B.** Venn diagrams illustrating GO terms that are commonly or uniquely regulated (FDR <0.05) in monocytes, CD4 T cells, CD8 T cells, B cells, and NK cells; due to infection alone, DI treatment of infected cells, or DI treatment of uninfected cells. Go terms that are enriched/depleted in all cell types by infection and DI treatment are printed in bold. **C.** GO term enrichment analysis of CD4 T cells, CD8 T cells, B cells, and NK cells.

anti-IFN effects similar to DI. Overall, the differences in anti-IFN activity of the three compounds correlated well with their ability to reduce STAT1 phosphorylation, as itaconate was clearly the weakest P-STAT1 inhibitor. The differential effects on AKT phosphorylation are particularly noteworthy, as they may begin to explain why itaconate and DI exert opposite effects on some mediators of inflammation. Itaconate did not reduce expression of *IFNB1*, *TNF* mRNA or IL1β protein in PBMCs. The lack of *TNF* repression by itaconate in that model is consistent with previous reports that it usually does not affect TNF-α levels [12]. Recent work has also revealed the potential of itaconate to actually enhance IL1-β release [41]. Of note, at higher doses, itaconate and strong electrophilic compounds can actually increase inflammation due to CASP-8 dependent inflammasome activation, and this effect is magnified when the cells are not pre-incubated with the compound before adding the pro-inflammatory stimulus [41], as we did in the PBMC infections. However, these authors did not investigate effects on IFN suppression. 4OI demonstrated the expected potent anti-inflammatory effects, but also broad effects on processes relating to, e.g., chromatin structure, the potential effects of which on cell homeostasis and cell-pathogen interactions will require further study. Taken together, these results re-inforce the notion that itaconate derivatives may take on quite different immunomodulatory properties than the native form, which may differ depending on the nature of the chemical modification.

## Anti-inflammatory effect of ectopic synthesis of itaconate in A549 cells

Despite the multitude of studies on effects of exogenous administration of itaconate derivatives on cellular inflammation, it had not been tested whether ectopic synthesis of itaconate in a cell type that does not naturally express *ACOD1* has similar effects. Our transfection study (**Fig 5**) demonstrated that ectopic, endogenously synthesized itaconate exerts very similar anti-IFN effects as exogenously added itaconate or DI. Of note, the achieved itaconate concentration is close to that measured in LPS/IFNγ stimulated dTHP-1 cells, i.e. 500 μM [42], suggesting that it is physiologically relevant.

## Cell-type specific and -independent anti-inflammatory effects of itaconate and DI

A particular strength of our study lies in the use of scRNAseq to identify major target cells in peripheral blood. Firstly, this analysis identified monocytes as the major PBMC type that is infected by IAV. Secondly, we found that on one hand DI can attenuate IFN-responses in lymphocytes and NK cell as well, but that the immunosuppressive effect is greatest on monocytes. Taken together, these results suggest that even though pharmacological use of itaconate and derivatives would modulate host responses in a diverse array of cells, monocytes/macrophages likely constitute a major functional hub in the genesis of inflammation in IAV infection and subsequent treatment responses to itaconate-based adjunct treatments.

## Antiviral effects

The strong reduction of viral titers by 4OI, but only weak reduction by DI, in A549 cells suggests that this effect is not mediated primarily by 4OI's electrophilic properties, as DI is a

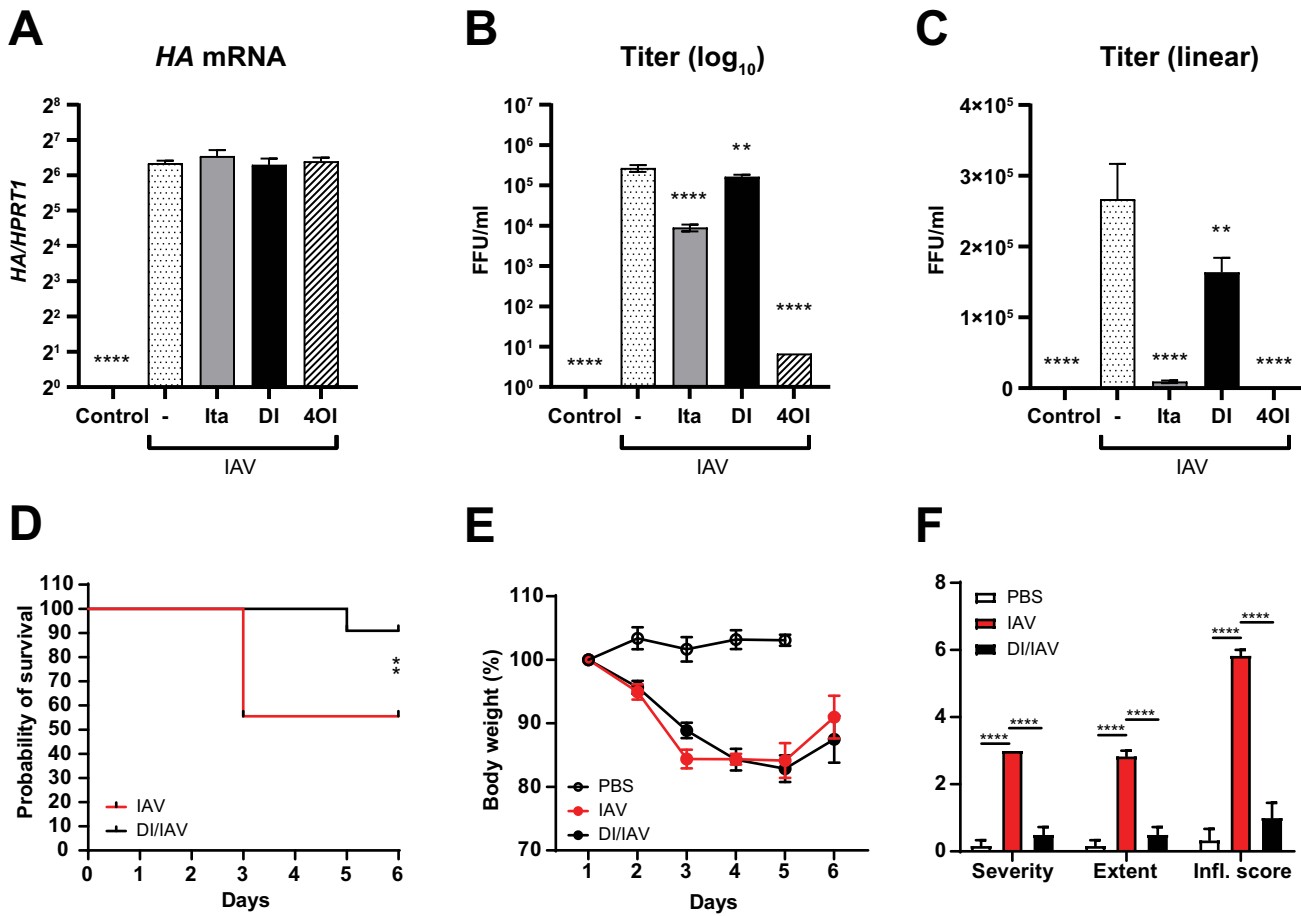

**Fig 11. A-C. Itaconates differ greatly in the ability to reduce progeny virions in productive IAV infection.** A549 cells were infected with IAV PR8M (MOI = 1) and treated with itaconate (20 mM), DI (0.5 mM) or 4OI (125 μM) as indicated. HA mRNA and viral titers were measured 24 h p.i. **A.** HA mRNA (RT-qPCR). **B,C.** A549 cells, titers in culture supernatants (foci-forming assay) 24 h p.i. expressed on $\log_{10}$ (B) or linear (C) scale. **D-F. DI treatment increases survival and prevents pulmonary inflammation in the mouse model of IAV infection.** 6–8 week-old female C57Bl/6J mice were infected with $1 \times 10^3$ FFU of mouse-adapted IAV (A/PR/8/34) and injected once daily with DI (50 mg/Kg intraperitoneally) for 5 days (n = 9), the first dose being given the day before infection, or PBS (n = 11). Body weight was determined daily and mice were sacrificed 5 d p.i. (PBS mock infection) or 6 d p. i. (IAV infection). **D.** Survival curve indicating higher survival in the DI-treated group ($p = 0.0097$, 2-tailed $t$ test). **E.** Weight loss curves indicating no effect of DI treatment on body weight. **F.** Pulmonary inflammation scores (H&E stained sections; same score as used in Fig 1) in randomly selected samples (n = 6 per group). DI treatment prevented pulmonary inflammation nearly completely. Representative histological findings are shown in **S25 Fig**. Viral titers in lung homogenates ($7.0 \times 10^6$ vs. $8.9 \times 10^6$ FFU) did not differ significantly between treated and untreated IAV-infected mice ($p = 0.8$, two-tailed $t$ test, n = 3 randomly selected samples per group).

stronger electrophile [18]. Likewise, even though it has been suggested that SDH inhibition mediates antiviral effects of itaconates [23], SDH inhibition does not explain the strong antiviral effect by 4OI as it does not inhibit this enzyme [18]. The lack of effect on viral RNA levels in A549 cells suggests inhibition at the posttranscriptional level. Indeed, Sethy et al. have reported inhibition of nuclear-cytoplasmic export of IAV ribonucleoprotein by compounds containing an itaconate-like center and have proposed the nuclear exportin chromosome region maintenance 1 (CRM1) as their cellular target [43]. Olagnier et al. recently reported that NRF2 activators including 4OI limit inflammation in cellular models of SARS-CoV-2 infection in an NRF2-dependent manner [25]. These authors in addition observed that 4OI inhibited infectivity of a variety of viruses, which proved to be independent of NRF2- and type

**Table 1. Comparison of key features of the three itaconates.**

| Parameter | Itaconate | Dimethyl itaconate | 4-Octyl itaconate |
|---|---|---|---|
| Reduction of type I IFN expression | ++ | +++ | +++ |
| Reduction of P-STAT1 | + | + | +++ |
| Effect on P-AKT | Up | Down | Down |
| Reduction of *ACOD1* mRNA (dTHP1) | + | ++ | +++ |
| Reduction of *ACOD1* mRNA (PBMCs) | — | ++ | +++ |
| Reduction of ROS | +++ | +++ | +++ |
| Reduction of IAV titers (A549) | ++ | + | +++ |
| Reduction of HA mRNA (PBMCs) | — | — | ++ |

Parameters were graded on the scale +, ++, and +++ to indicate estimated differences among the three compounds.— = no effect detected.

HA = hemagglutinin

ROS = reactive oxygen species

I IFN-signaling, but influenza viruses were not tested. Our finding of a strong suppression of virion production in A549 cells and IAV RNA replication in PBMCs lends further support to the notion that 4OI has antiviral effects which may be mediated by more than one mechanism. The gene expression analysis of itaconate, DI- and 4OI-treated IAV-infected PBMCs identified several processes/pathways uniquely regulated by 4OI. Further research should focus on assessing whether any of them could contribute to its antiviral potential.

## Infection-independent effects on target cells

By using relatively high doses of itaconate and DI and assessing effects of the compounds on uninfected cells or their effects on processes that were not affected by infection alone, our study also provides a first comprehensive look at global effects of the compounds on target cells. In particular, itaconate and 4OI exerted pronounced, broad effects on gene expression in A549 cells and PBMCs, respectively. Clearly, it will be of great interest to investigate how these effects relate to differences in toxicities on different cell types, or to differences in their effects on virus-host interactions.

## Importance of endogenous ACOD1 expression

In the mouse model, we observed a more severe clinical phenotype and higher inflammation in *Acod1*[-/-] mice. This agrees with work in a mouse model of *M. tuberculosis* infection, where inflammation and mortality were higher in the *Acod1*[-/-] animals [26]. However, IAV infection was also assessed in that study, and deleting the *Acod1* gene did not affect weight loss or survival. Those authors used equal numbers of male and female mice, whereas we used female mice only. Further research is necessary into possible sex-specific effects of the *Acod1* gene in influenza infection. Of note, we found upregulation of *ACOD1* mRNA by IAV infection in myeloid cells and in blood from patients with moderate and severe influenza infection. Thus, induction of *ACOD1* expression (and thereby itaconate synthesis) likely is a common feature in human influenza. We have recently identified the active center of the ACOD1 enzyme and found that naturally occurring loss-of-function mutations are extremely rare, suggesting that physiologic itaconate responses have been important in human evolution (6). It is therefore tempting to speculate that one beneficial role of an active ACOD1/itaconate axis is to reduce the risk of death or organ damage from infection-associated inflammation and that loss-of-function genotypes have been selected against during times of high infectious disease burdens. In human PBMCs, we observed unchanged *ACOD1* and *IFNB* mRNA expression after

itaconate treatment, but lower *ACOD1* expression and strong anti-IFN effects after treatment of dTHP1 cells. Moreover, reduction of *ACOD1* expression by itaconate, DI, and 4OI correlated with their ability to inhibit STAT1 phosphorylation. One might therefore postulate that a reduction of *ACOD1* expression is a requirement for therapeutic efficacy of treatment with itaconates. However, our data comparing BMDMs from WT and *Acod1*$^{-/-}$ mice strongly suggest that ACOD1 activity is not required for an anti-inflammatory response to treatment, at least with DI. Considering the well-established role of IFN signaling in regulating *ACOD1* expression [9, 44], reduced *ACOD1* expression by the treatments may, therefore, be mostly an epiphenomenon of diminished IFN signaling.

### Therapeutic effect in the mouse model of IAV infection

Consistent with the reduction of two key pro-inflammatory chemokines (CXCL10 and CCL2), seen in the cellular models, DI exerted broad anti-inflammatory effects to the point that lungs from treated animals were nearly indistinguishable from lungs from control mice. Its lack of effect on weight loss was unexpected and clearly suggests that not all other infection-associated factors normalized or (consistent with the induction of several cytokines by DI in uninfected dTHP1 cells) that some deleterious factors may have been induced. The latter would be consistent with our observation that DI treatment induced release of several cytokines from uninfected dTHP1 cells. Our preliminary results that viral titers were grossly unaffected in this model agree with the only modest anti-viral effect of DI seen in A549 infection. Additional studies will be necessary to verify these observations and to elucidate the underlying mechanisms.

### Clinical implications

The current COVID-19 pandemic has, once again, focused the research limelight on the deleterious effects of overshooting innate immune responses in acute viral infections. In a current report, Lee et al. found that intense IFN responses in PBMCs from patients are a hallmark of both severe COVID-19 and influenza [45]. Of note, a recent systematic review of 45 studies on corticosteroid use for COVID-19 found a significant beneficial effect [46], whereas to date there are no clinically effective antivirals against SARS-CoV-2. Therefore, the concept of adjunct treatments to control deleterious host immune responses in viral infections is more pertinent than ever. Considering their combined cytoprotective, anti-inflammatory, and antiviral effects, itaconates constitute a highly promising class of compounds for such applications. Considering the lower required concentrations of DI and 4OI, but also the striking anti-viral effects of 4OI, it appears that chemical variants of itaconate (particularly those related to 4OI), rather than its native form, will take the lead in further translational development to the bedside.

### Limitations of the present work

Since a substantial part of our work is based on data of *ACOD1* mRNA expression, rather than itaconate measurements, we cannot be certain that the observed differences in ACOD1 expression always translate into biologically meaningful differences in itaconate concentration. Our work demonstrated strong anti-inflammatory effects of DI in the mouse model of IAV infection. The nearly complete prevention of pulmonary inflammation by DI agreed well with its ability to downregulate both CXCL10 and CCL2 (which has also been described by Wang et al, [47]. However, we did not attempt to characterize more subtle immunomodulatory effects of DI on the infected lung, e.g. changes in the composition of lung immune cell or cytokine populations, which might be seen better at lower DI doses. In spite of its strong anti-inflammatory

effect, DI treatment did not affect weight loss of the infected mice. This puzzling observation suggests that DI may not ameliorate circulating levels of factors that mediate weight loss in this model. Considering that viral titers were not reduced, weight loss is likely driven by ongoing viral replication. We are intrigued by the observation that *ACOD1* transcription was more strongly induced in M2 than in M1 polarized macrophages. This should be followed up in greater depth, e.g. by fine-mapping *Acod1* mRNA expression in pulmonary macrophage subsets and by comparing relative abundances of M1 and M2 macrophages in IAV-infected WT and *Acod1*[-/-] mice. The increased susceptibility of *Acod1*[-/-] mice to IAV infection also requires further clarification, particularly regarding the relative contribution of viral replication and dysfunctional anti-inflammatory mechanisms to the observed higher pulmonary inflammation and disease severity.

## Methods

### Ethics statements

The study was approved by the Ethics Committee of Hannover Medical School (MHH), and all donors gave informed consent for experimental use of their tissue.

Animal procedures were performed at the Helmholtz Centre for Infection Research, Braunschweig (Germany) following guidelines from the Federation of European Laboratory Animal Science Associations. The study was approved by the regulatory authority of the German Federal Sate of Lower Saxony (Niedersächsisches Landesamt für Verbraucherschutz und Umwelt, LAVES, permit no. 33.4-42502-04-13/1281).

### Cell culture

Cell lines were obtained from German Collection of Microorganisms and Cell Cultures GmbH (DSMZ), Braunschweig, Germany. The human myelomonocytic leukemia cell line THP-1 was propagated in RPMI 1640 medium (GIBCO Life Technologies) supplemented with 10% fetal calf serum (FCS) and 2 mM L-glutamine. Cells ($2.5 \times 10^5$ cells/ml) were differentiated with 200 nM phorbol-12-myristate-13-acetate (PMA, Sigma-Aldrich, product no.P-8139) for 48 h, followed by incubation in fresh RPMI medium for another 24 hours (h). Human adenocarcinoma cells resembling type II alveolar epithelial cells (A549) were propagated in DMEM medium supplemented with 10% FCS and 2 mM L-glutamine.

### Overexpression of ACOD1

A549 cells ($2 \times 10^5$) were transfected with the plasmid pCMV6-Hu-ACOD1 (5 µg, Human ACOD1 transfection-ready DNA from OriGene, which was described by Michelucci and his colleagues in 2013 [5]) or empty pCMV6 plasmid as negative control. Transfections were done by using Lipofectamine LTX and PLUS reagent (Invitrogen) and then incubated for 24 h and 48 h.

### Primary cell isolation and differentiation

Primary human monocytes were isolated from buffy coats from healthy blood donors by density gradient centrifugation (Histopaque, Sigma-Aldrich). For studies of monocytes and M1/M2-type macrophages, CD14+ cells were isolated by magnetic activated cell sorting (CD14 + Cell Isolation Kit; Miltenyi Biotec). $2 \times 10^5$ monocytes were then differentiated into M1-type macrophages by incubation with 1000 U/ml GM-CSF (granulocyte macrophage-colony stimulating factor, CellGenix) and into M2-type macrophages by incubation with 100 ng/ml M-CSF

(macrophage-colony stimulating factor, Miltenyi Biotec) in serum-free DC medium (Cell-Genix) for 5 days.

## Virus strains

A common reference strain of influenza A virus (A/PuertoRico/8/34 [H1N1], referred to as PR8M), kindly provided by Stefan Ludwig (University of Münster, Germany) [48] was used throughout. In the experiment shown in **Fig 2**, we additionally used the clinical isolate Gi-WT (A/Giessen/6/2009 H1N1-WT) and the reassortant Gi-NS-PR8 containing the NS segment of the PR8 strain on the backbone of Gi-WT, which replicates faster than the WT [34]. Viruses were propagated in the chorio-allantoic cavity of 10-day-old embryonated eggs for 48 h at 37°C. Fluid from the chorio-allantoic cavity was collected and the virus was titrated by focus-forming unit (FFU) assay and stored in aliquots at -80°C until use.

## Viral infections and treatment with itaconate, DI, and 4OI

$2.5 \times 10^5$ dTHP-1 or A549 cells were infected with the indicated IAV strain at an MOI of 1, cells were centrifuged at 300 g for 15 min and incubated at 37°C for 2 h. Cells were subsequently washed twice with PBS and fresh post-infection medium was added to the cells. Cells were re-incubated at 37°C for the indicated lengths of time. The respective amount of itaconate, DI, or 4OI was added to RPMI/DMEM complete medium and pH was adjusted to 7.5 with 1 M KOH solution. 4OI stock (100 mM) was prepared in DMSO, and a final concentration of 4OI of 25 µM thus resulted in a 0.025% concentration of DMSO in the media. Media were vacuum-filtered (0.22 µm pore size, Millipore). Unless stated otherwise, cells were incubated overnight with the compounds, and virus was then added for 2 h. Unbound virus was then removed by replacing the medium with fresh medium containing the compounds at the same concentrations. For PBMC infections, cells were isolated from 7 donors and seeded as $4 \times 10^6$ cells/well in a 6 well plate and simultaneously infected for 12 h with strain PR8M (MOI = 1) with or without itaconate (10 mM), DI (0.5 mM), or 4OI (25 µM). Thus, PBMCs were not preincubated with the compounds before infection. Treatment of human lung tissue was also performed without preincubation (see below).

## Mouse model of IAV infection (**Fig 1**)

*Acod1*$^{-/-}$, *Acod1*$^{+/-}$, and *Acod1*$^{+/+}$ mice with C57BL/6N background had originally been generated by Dr. Haruhiko Koseki in the RIKEN Institute (Yokohama, Japan) by the use of stem cells which were purchased from the Repository of Knockout Mouse Project under strain ID Irg1tm1a (KOMP) Wtsi. Age-matched groups of female *Acod1*$^{-/-}$, *Acod1*$^{+/-}$, and *Acod1*$^{+/+}$ mice were derived from the same heterozygous breeding pair. Animals were bred at the University of Luxembourg and then transferred to the Animal Experimental Unit of the Helmholtz-Centre for Infection Research, where they were kept under Animal Biosafety Level 2 and non-specific pathogen-free conditions. Animals were housed separated according to genotype, up to 5 mice per cage. Mice were sedated by intraperitoneal injection of ketamine (10 mg/mL) and xylazine (1 mg/mL) in 0.9% NaCl, infected intranasally with $5 \times 10^5$ FFU (20 µL) IAV (strain A/California/7/2009 [H1N1], reagent 15/252, obtained from National Institute for Biological Standards and Control, London, UK) and then observed for weight loss and survival for up to 15 days. Mice with weight loss >20% had to be sacrificed ($CO_2$ asphyxiation) and were counted as non-survivors. In a separate experiment, mice were sacrificed on days 8/9 and 14 p.i. (one mouse that died on day 11 was also included) and lungs examined for histopathological changes. Formalin-fixed, paraffin-embedded sections of mouse lungs were stained with hematoxylin/eosin (H&E). To assess the degree of inflammation and histopathological lesions,

**Table 2. Scoring system to assess inflammation in IAV-infected mouse lung.**

| Parameter | 0 | 1 | 2 | 3 |
|---|---|---|---|---|
| Severity of infiltration of inflammatory cells | Not present | Mild | Moderate | Severe |
| Extent (area involved) | No changes | 1–30% | 30–70% | >70% |

The total inflammation score is the sum of the two component scores, thus ranging from 0 to 6.

severity and extent of inflammation (Table 2) were scored by an expert in mouse pathology (MP), who was not aware of the identity of the slides, according to the scale below.

### Treatment of IAV infection mouse model with DI (Fig 11)

Infections were performed at the National Research Centre, Cairo (Egypt), as previously described [49], following guidelines from the Federation of European Laboratory Animal Science Associations. The study was approved by the animal research committee of the National Research Centre, Cairo (Egypt).

Briefly, female 6–8 week-old C57BL/6J mice were anesthetized by intraperitoneal injection with ketamine/xylazine, infected intranasally with $1\times10^3$ FFU of influenza virus (A/PR/8/34), and were injected intraperitoneally once daily for 5 days with DI (50 mg/Kg dissolved in PBS) or PBS only. The first dose was administered the day before infection. Body weight and survival were assessed daily. Mice were treated for 5 days, and surviving mice were sacrificed on day 6 p.i. Lungs were divided into two parts, where one half was homogenized for measurement of viral titers using the tissue culture infective dose (TCID50) method on MDCK cells and the other half was fixed in formaldehyde for histopathological processing. The experiment was performed twice with 5–6 mice per group, and final analyses were performed using the pooled sample of 9–11 mice per group.

### Preparation of bone marrow derived macrophages (BMDM)

Femurs were prepared from WT or *Acod1*$^{-/-}$ C57BL/6N mice, washed with 70% EtOH and then kept in PBS on ice. For isolation of immune cells, the bones were placed in a mortar with 5 ml PBS + 2% (v/v) FBS. The bones were then crushed with a pestle and the cell suspension was filtered through a 100 µl cell strainer into a 50 ml reaction tube. The bones were ground 3 times in 5 ml PBS + 2% (v/v) FBS. After that the cell solution was centrifuged for 12 min at 250 xg at 4˚C. The cell pellet was resuspended in 2 ml ACK buffer to lyse erythrocytes. Lysis was stopped after 2 min by adding at least 5 ml FACS buffer. Next, the cell suspension was centrifuged for 10 min at 250 xg and 4˚C. Following centrifugation, the pellet was resuspended in 1 ml PBS and filtered through a 70 µl cell strainer. The cell count was determined using the Countess system (Invitrogen, Carlsbad, CA, USA). For *in vitro* differentiation of monocytes, 12 well cell culture plates were used. Per well, $0.3 \times 10^6$ cells were seeded into 1 ml differentiation medium (RPMI 1640, 10% FBS, 20% supernatant from the M-CSF-secreting cell line L929). After two days of cultivation, 1 ml fresh differentiation medium was added. On day 5, medium was aspirated, cells were washed with 2 ml warm PBS, and fresh differentiation medium was added. On day 7, the cells were pre-incubated with 0.25 mM DI for 6 h, then incubated for 2 h with IAV PR8 (MOI = 1, starting from the original 300,000 cells) or with 15 ng/ml IFNγ. Cells were then incubated for 12 h with fresh medium alone or medium containing 0.25 mM DI. Expression of *Cxlc10* and *Actb* mRNA was determined by RT-qPCR using the primers listed in Table 3.

**Table 3. List of RT-qPCR primers.**

| Gene | Primer name | Sequence |
|---|---|---|
| ISG15 | ISG15-F | TGTCGGTGTCAGAGCTGAAG |
| | ISG15-R | AGAGGTTCGTCGCATTTGTC |
| IL6 | IL6-F | CTACATTTGCCGAAGAGCCC |
| | IL6-R | CCCTGACCCAACCACAAATG |
| HPRT | HPRT-F | GAACGTCTTGCTCGAGATGTG |
| | HPRT-R | CCAGCAGGTCAGCAAAGAATT |
| IL1B | IL-1β-F | TACCCAAAGAAGAAGATGGAA |
| | IL-1β-R | GAGGTGCTGATGTACCAGTTG |
| CXCL10 | CXCL10_F | CTGCTTTGGGGGTTTATCAGA |
| | CXCL10_R | CCACTGAAAGAATTTGGGC |
| A20 | A20_F | ATGCACCGATACACACTGGA |
| | A20_R | CACAAGCTTCCGGACTTCTC |
| IL10 | IL10-F | TACCTGGGTTGCCAAGCCT |
| | IL10-R | AGAAATCGATGACAGCGCC |
| HA | HA-F | CTCGTGCTATGGGGCATTCA |
| | HA-R | TTGCAATCGTGGACTGGTGT |
| IFNB1 | IFNβ_F | CAGCAATTTTCAGTGTCAGAAGC |
| | IFNβ_R | TCATCCTGTCCTTGAGGCAGT |
| ACOD1 | ACOD1_F | ATGCTGCTTTTGTGAACGGTG |
| | ACOD1_R | CTACCACGGAAGGGGGATGGA |
| Cxcl10 (mouse) | mCXCL10_3' | TCTCACTGGCCCGTCATC |
| | mCXCL10_5' | GCTGCCGTCATTTTCTGC |
| Actb (mouse) | mActβ_3' | TCCTTGTGACCCATTCCCA |
| | mActβ_5' | CTTCTTTGCAGCTCCTTCGT |

## Human lung tissue explant model

Tissue was obtained at the time of medically indicated lung transplantation from patients with end-stage lung disease due to emphysema or pulmonary arterial hypertension. The overall tissue quality and gross pathological changes were assessed by a board-certified pathologist (Dept. of Pathology, Hannover Medical School). Healthy appearing tissue was then dissected and sectioned into pieces of approx. 30–50 mm$^3$ (typical weight, 30–50 mg). Tissue pieces were cultured overnight in individual wells in RPMI medium, containing DI (1 mM) or itaconate (25 mM) as indicated. After this pre-treatment + washing step in medium, tissue pieces were infected with IAV ($2 \times 10^5$ FFU/ml) for 24 h at 37˚C in medium containing itaconate or DI.

## RT-qPCR

RNA was purified using the Nucleospin RNA purification kit (Machery Nagel) and on-column removal of DNA by digestion with rDNase (Machery Nagel) for 15 min. at RT. cDNA was synthesized with the PrimeScript kit (TaKaRa, Shiga, Japan) using 400 ng RNA in a 10 µl reaction. RT-qPCR reactions were set up in a final volume of 20 µl, using the SensiFast SYBR No-ROX Kit (Bioline, Taunton, MA) and the primers listed in Table 3. RT-qPCR was performed in a LightCycler 2.0 instrument (Roche, Mannheim, Germany), using 45 cycles of the following program: 95˚C for 15 sec., 60˚C for 15 sec., and 72˚C for 15 sec. To exclude artefacts resulting from primer dimer formation, melting curve analysis was performed using the sequence 95˚C for 15 sec., 60˚C for 15 sec., 95˚C for 1 min. and 37˚C for 30 sec. Relative expression of the

host mRNA targets was calculated using the $2^{-\Delta\Delta CT}$ method [50], and expression of IAV *HA* mRNA by normalizing against expression of *HPRT* as internal control.

## Microarray analyses

RNA was extracted using the RNeasy kit (Qiagen), RNA quality was checked with a Bioanalyzer 2100 (Agilent Technologies) and samples with RNA integrity number >7 were used for microarray analysis. Total RNA/Poly-A RNA Control Mixture was prepared by adding poly-A RNA controls (Affymetrix). This poly-A RNA was then used to prepare double standard cDNA. Antisense RNA (complimentary RNA or cRNA) was then synthesized and amplified by *in vitro* transcription (IVT) of the ds-cDNA template using T7 RNA polymerase. Enzymes, salts, inorganic phosphates, and unincorporated nucleotides were removed to prepare the cRNA for 2nd-cycle ds-cDNA synthesis. After verifying quality and yield of cRNA, sense-strand cDNA was synthesized by the reverse transcription of cRNA. cRNA template was hydrolyzed leaving ds-cDNA, which was then purified. cDNA was fragmented and labeled by terminal deoxynucleotidyl transferase (TdT) using the Affymetrix proprietary DNA Labeling Reagent that is covalently linked to biotin. Cartridge Array Hybridization was conducted on the GeneChip Instrument (Affymetrix) using Thermo Fisher Scientific microarrays (Clariom S Assay, human). Raw.CEL files from Clariom S Pico Assay microarrays for hg.38 were imported into the Transcriptome Analysis Console (TAC4.0.2) Software (ThermoFisher Scientific). Array QC and data normalization was performed using the robust multiarray average (RMA) method[51]. Differentially expressed genes (DEGs) were identified using limma:Linear Models for microarray and RNA-Seq Data (www.Bioconductor.org) and one-way ANOVA.

## Differential gene expression analysis of bulk RNAseq data

Differential gene expression for **Fig 1B** was calculated using generalized linear models contained in the edgeR package (doi:10.1093/bioinformatics/btp616). P values were adjusted using false discovery rate (FDR), using FDR <0.05 to define significance of differential gene expression.

## Single cell RNA sequencing

**Barcoding and cDNA synthesis.** The single cell suspension was loaded onto a well on a 10x Chromium Single Cell instrument (10x Genomics). Barcoding and cDNA synthesis were performed according to the manufacturer's instructions. Briefly, the 10x GemCode Technology partitions thousands of cells into nanoliter-scale Gel Bead-In-EMulsions (GEMs), where all the cDNA generated from an individual cell share a common 10x Barcode. In order to identify the PCR duplicates, Unique Molecular Identifier (UMI) was also added. The GEMs were incubated with enzymes to produce full length cDNA, which was then amplified by PCR to generate enough quantity for library construction. Quality was checked using the Agilent Bioanalyzer High Sensitivity Assay.

**Library construction and quality control.** The cDNA libraries were constructed using the 10x Chromium Single cell 3' Library Kit according to the manufacturer's original protocol. Briefly, after the cDNA amplification, enzymatic fragmentation and size selection were performed using SPRI select reagent (Beckman Coulter, Cat# B23317) to optimize the cDNA size. P5, P7, a sample index and read 2 (R2) primer sequence were added by end repair, A-tailing, adaptor ligation and sample-index PCR. The final single cell 3' library contains a standard Illumina paired-end constructs (P5 and P7), Read 1 (R1) primer sequence, 16 bp 10x barcode, 10 bp randomer, 89 bp cDNA fragments, R2 primer sequence and 8 bp sample index. For post

library construction QC, 1 μl of sample was diluted 1:10 and ran on the Agilent Bioanalyzer High Sensitivity chip for qualitative analysis.

**Single-cell RNA sequencing and generation of data matrix.** PBMC were infected and treated as for the microarray analyses. Cells were then stained with propidium iodide (Apoptosis Detection Kit, eBioscience, cat. no. 88-8005-74), and dead cells and doublets were removed by FACS. Live cells (including those undergoing apoptosis) were used to make single-cell suspensions using the 10x Genomics platform. Libraries were sequenced on an Illumina NovaSeq 6000 2x50 paired-end kits using the following read length: 26 bp Read1 for cell barcode and UMI, 8 bp I7 index for sample index and 89 bp Read2 for transcript. Cell Ranger 1.3 (http://10xgenomics.com) was used to process Chromium single-cell 3' RNA-seq output. First, "cellranger mkfastq" demultiplexed the sequencing samples based on the 8bp sample index read to generate fastq files for the Read1 and Read2, followed by extraction of 16bp cell barcode and 10bp UMI. Second, "cellranger count" aligned the Read2 to the pre-built human reference genome (hg19, GRCh38) using STAR. Then, aligned reads were used to generate data matrix only when they had valid barcodes and UMI, and mapped to exons (Ensembl GTFs GRCm38. p4) without PCR duplicates. Valid cell barcodes were defined based on UMI distribution.

*Single-cell gene expression data analysis including filtering, normalization, and clustering* was processed using Seurat V3.1 (https://satijalab.org/seurat/) in the R environment (www.bioconductor). Cells were removed when they had $< 500$ genes and $> 25\%$ reads mapped to mitochondrial expression genome. After filtering and doublet removal, 6456 single cells (control = 1870, DI = 1474, IAV = 1701, IAV & DI = 1411) remained. Raw counts were normalized by the global-scaling normalization method "LogNormalize". Principal component analysis score was used to determine the 'dimensionality' of all cells. Scatter plots were obtained using the UMAP method. Cell clusters were identified as different cell populations based on the expression of recently published canonical markers (**Table 4**). Differential gene expression was determined for each cell type using DESeq2 [52]. IAV infected cells were identified by the presence of IAV encoded transcripts, as mapped against NIAID Influenza Research Database [53], www.fludb.org/brc/fluStrainDetails.spg?strainName=A/Giessen/6/2009(H1N1)&decorator=influenza. mRNA encoding the viral M protein could not be identified, likely due to a sequence mismatch.

## Enzyme-linked immunoassay

Supernatants from the same PBMC as used for RNA analyses were collected 12 h p.i. Protein concentrations were measured using ELISA MAX Deluxe Sets (Biolegend; IL-1β cat. no. 437004; IL-6, 430501; TNF-α, 430201; IFN-γ, 430101) as per the manufacturer's instructions. Briefly, capture antibody was used to pre-coat the wells O/N. After removing the capture

**Table 4. mRNA markers used to define PBMC sub-populations in scRNAseq.**

| Cell type | Sub-type | Marker(s) | Sub-sub type | Marker(s) | Ref. |
|---|---|---|---|---|---|
| T cells | CD4 | *CD3D, IL7R* | CD4 naïve | *LEF1, ATM, SELL, KLF2, ITGA6* | [54–56] |
| | CD4 | | Effector memory | *KLRB1 IL7R* | [55, 56] |
| | CD4 | | CD4 memory | *PASK* | [55, 56] |
| | CD4 | | Treg | *FOXP3* | [57] |
| | CD8 | *CD3D, CD8A* | Effector memory | *GZMK* | [56, 58] |
| | CD8 | | Cytotoxic | *GNLY* | [54, 56] |
| B cells | | *MS4A1* | Activated B cells | *MIR155HG* | [56] |
| NK cells | | *NKG7 GNLY* | | | [56] |
| Monocytes | | *CD14, LYZ* | | | [56] |

antibody and washing the plates, block buffer was applied for 1 h at room temperature at 500 rpm. Again, plates were washed at least 4 times before adding the samples (100 μL, diluted as needed) and standards for 2 h at RT. The plates were then washed again 4 times, and detection antibody was added to each well. The plate was further incubated for 1 h at RT and then washed 4 times. Avidin-HRP conjugate was added and incubated for 30 min. Substrate was added until color developed, and the samples were measured after 15–20 min at 450 and 570 nm.

## Multiplex assay for inflammation-related polypeptides

Concentration of 27 inflammation-related human proteins were measured in cell culture supernatants by using Human 27-plex BIORAD cytokine panel (Cat. No.171-A1112, BIORAD). Standard curves were generated for all 27 cytokine standards with eight 2-fold serial dilutions starting from 32 ng/ml. 100 μl of assay buffer was added to each well of a microfiltration plate, followed by addition of 50 μl of beads suspension. After washing the beads with assay buffer, 50 μl of standard or the sample (supernatant) was added to each well and incubated for 30 min at room temperature with gentle shaking. 25 μl of antibody premix for detection was added into each well followed by incubation for 30 min at room temperature with gentle shaking. Three washing steps were performed using 50 μl of assay buffer in each well, followed by washing with streptavidin solution for 10 min at room temperature with shaking. Final washing with 125 μl of assay buffer was performed before quantification by using BIO-PLEX Manager software version 4.

## Immunofluorescense

THP-1 cells were seeded in a 24 well plate with coverslips and differentiated at 80% confluency. Cells were fixed with 2.5% formaldehyde for 20 min and the reaction stopped by adding 0.1 M glycine for 5 minutes (min). Cells were then permeabilized with 0.2% Triton X100 in PBS for 5 min. Cells were incubated for 1 h with primary antibody (rabbit-anti p65 mAB, Cell Signaling, cat. no. 8242S) 1:200 diluted in BSA/PBS 1 mg/ml. Cells were then washed three times with PBS and incubated in the dark for 45 min with Alexafluor594 secondary antibody (1:100) (Cell Signaling, cat.# 8889S) in BSA/PBS 1 mg/ml and incubated in the dark for 45 min. Cells were washed three times with PBS. To stain nuclei, DAPI was added at 1:10,000 in PBS for 10 min. Cells were washed three times with PBS and mounted with 20 μl of DAKO mounting medium per coverslip on glass slides and stored in the dark until imaged. The images were analysed using Image J s/w (imagej.net).

## Quantification of itaconate

Itaconate concentrations were measured by liquid-chromatography tandem mass-spectrometry (HPLC-MS/MS) using a liquid chromatography system (NEXERA LC, Shimadzu, Japan) coupled to a triple quadrupole—ion trap mass spectrometer (QTRAP 5500, Sciex, Framingham, MA) and Analyst 1.7 software (Sciex, Framingham, MA, USA), as described in detail in [59].

## Statistics

Data are shown mean ± standard error of the mean (SEM) unless stated otherwise. Significance of between-group and across-group differences was assessed with the statistical tests indicated in the figure legends, using Prism 5.02 and 8.0 (GraphPad Software Inc.). Wherever exact p values are not stated, significance is graded as $^{*}p < 0.05$, $^{**}p < 0.01$, $^{***}p < 0.001$.

Principle component analysis was performed with the R package "PCAtools" (https/github.com/kevinblighe/PCAtools), and **hierarchical clustering analysis** with the R tool pheatmap (cran.r-project.org/web/packages/pheatmap/pheatmap.pdf).

### Reanalysis of published data sets

Reanalyses of publically available published data sets were performed in the R environment (version 3.6.3) (R_Core_Team, 2014), using package beeswarm (version 0.2.3.) (Eklund, 2016) for visualization as indicated.

### Gene set enrichment analysis

Features with linear fold change (FC) $\geq |1.5|$ and unadjusted P-value $\leq 0.05$ were used for enrichment analysis [60] based on gene ontology (GO Biological Process) and KEGG pathways, using GEne SeT AnaLysis Toolkit [61]. Enrichment/depletion strip charts were made using ggplot2 [62]. Volcano plots to show DEGs were made using R package "EnhancedVolcano" (https://github.com/kevinblighe/EnhancedVolcano). Molecular Signatures Database (MSigDB) v7.2 [63] was used to collect the information of genes involved in gene set enrichment analyses.

### Supporting information

**S1 Fig. IAV infection induces higher itaconate concentrations in lungs from DBA/2J than from C57BL/6J mice.** Mice were infected with IAV strain PR8M ($2x10^3$ FFU) as described in (31). Levels of HA and *Acod1* mRNA (RT-qPCR) and itaconate (LC-MS/MS) were measured on days 3 and 4 p.i. Lungs collected 3 days after mock infection with PBS were used as controls. Means ±SD. *p<0.05; **p<0.01 (*t* test). **A.** HA mRNA. **B.** *Acod1* mRNA. Reference for fold change = PBS control of the respective mouse strain. **C.** Itaconate.
(TIF)

**S2 Fig. Cytotoxicity data of itaconate, DI, and 4OI on dTHP1 and A549 cells.** Cytotoxicity was determined by applying increasing concentrations of the compounds to dTHP1 ($25 x10^3$ cells/well) and A549 cells ($20 x10^3$ cells/well) and measuring cell respiration with the 3-(4,5-dimethylthiazol-2-yl)-2,5-diphenyltetrazolium bromide (MTT) assay after 24 h. Curves for dTHP1 and A549 cells are shown in the same graphs for comparison, illustrating that itaconate and 4OI tend to be more toxic on dTHP1 cells, whereas DI is more toxic on A549 cells. **A.** Itaconate (n = 3 experiments, each with n = 8 replicates). **B.** DI (n = 3 experiments, each with n = 8 replicates). **C.** 4OI (n = 1 experiment with n = 8 replicates). Means ±SEM. **D.** Table summarizing the concentrations at which mean viability was reduced by 5, 10, 20 or 50%. The concentrations used in this study were selected to be < CC05. CC = concentration at which a given percentage of cells was non-viable.
(TIF)

**S3 Fig. Dose response curves of itaconate and DI treatment of IAV-infected dTHP-1 cells.** dTHP-1 cells were pretreated overnight with the indicated concentrations of itaconate or DI, infected with IAV (PR8M; MOI = 1), and expression of *HA* and *CXCL10* mRNA was measured after 12 h (RT-qPCR) (n = 3). Reference for fold change = uninfected, untreated 12 h.
(TIF)

**S4 Fig. The impact of itaconate and DI treatments on dTHP1 cell transcriptomes is greater than that of IAV infection.** dTHP1 cells were infected with IAV (PR8M, MOI = 1) and gene expression assessed by microarray analysis 12 h p.i. PCA based on the microarray analysis

shown in **Fig 3**. Untreated and treated IAV-infected cells cluster together, whereas untreated infected cells are clearly separated from all other groups in both principle components.
(TIF)

**S5 Fig. A. Hierarchical clustering analysis of effects of itaconate and DI on IAV-infected dTHP-1 cells.** dTHP1 cells were infected with IAV (PR8M, MOI = 1) and gene expression assessed by microarray analysis 12 h p.i. Analysis based on the same microarray data as used for **Figs 3** and S4, using the top 100 DEGs (FDR F-test <1.89E-09). Uninfected cells and untreated IAV-infected cells form one clade and itaconate and DI treated IAV-infected cells the other clade. From top to bottom the brackets identify the following clades: (i) a large clade of DEGs that are upregulated upon itaconate treatment, (ii) DEGs that are upregulated by both itaconate and DI, (iii) a predominantly pro-inflammatory clade that is induced in IAV infection and globally downregulated by both treatments, (iv) DEGs downregulated by both itaconate and DI, (v) DEGs downregulated only by itaconate, (vi) DEGs upregulated only by DI. **B. Enrichment plots based on the 99 genes contained in GO term *Response to type I interferon*,** based on the microarray data used for **Figs 3**, S4, and **S5A**. Enrichment scores are plotted on the y-axis, the mRNAs (identified by vertical lines) along the x-axis, ranked by fold change. The plots illustrate pronounced enrichment in a large number of genes due to infection, which is nearly quantitatively depleted by treatment with both itaconate and DI.
(TIF)

**S6 Fig. Itaconate, DI, and 4OI induce different potential recruitment programs in dTHP1 cells. A.** Supernatants of uninfected and IAV-infected (PR8M, MOI = 1) dTHP1 cells, with or without itaconate or DI treatment, were analyzed for concentrations of 27 cytokines/chemokines by multiplex microbead array 12 h after infection or mock treatment. A hierarchical clustering analysis was carried out with those targets that were detected above limit of detection of the assay in most samples. The microarray analysis of cellular gene expression in the same experiment is shown in **Fig 3** (n = 3). **B.** Supernatants of uninfected and IAV-infected (PR8M, MOI = 1) dTHP1 cells, with our without DI or 4OI treatment, were analyzed 12 h p.i. with the same 27-plex assay as in **A**. More targets were detected > LOD than in **A**, the effect of DI was, overall somewhat more pronounced, and DI treatment increased IL-10 levels in addition to IL-8 and IL-9. 4OI differs from DI in that it reduces levels of most targets more efficiently and does not increase IL-8, -9, and -10 levels.
(TIF)

**S7 Fig. Itaconate and DI treatments lead to a global downregulation of IFN-I signaling in IAV-infected dTHP-1 cells.** Cells were infected with IAV (PR8M, MOI = 1) and gene expression assessed by microarray analysis 12 h p.i. Enrichment analysis of gene ontology (GO Biological Process) terms based on the microarray data shown in **Fig 3**, using DEGs ($p<0.05$, FC >|1.5|) as input. GO terms are listed in reverse alphabetical order from top to bottom (page split after "protein"), with upright (red) triangles indicating enrichment (upregulation) and downward pointing (blue) triangles depletion (downregulation). Enriched GO terms (FDR ≤0.05) reveal major induction of antiviral and inflammatory responses by infection, which are decreased by both itaconate and DI treatment (most relevant inflammation and infection related GO terms are printed in bold font).
(TIF)

**S8 Fig. KEGG-pathway analysis of effects of itaconate and DI treatment on IAV-infected dTHP-1 cells.** dTHP1 cells were infected with IAV (PR8M, MOI = 1) and gene expression assessed by microarray analysis 12 h p.i. Analysis based on the microarray data set used for **Fig 4**. A broad dampening of IFN-related and pro-inflammatory pathways by both compounds is

evident.
(TIF)

**S9 Fig. Effects of increasing doses of itaconate and DI on *HA* and *CXCL10* expression in IAV-infected A549 cells.** RT-qPCR using *HPRT1* as reference. A549 cells were infected with IAV (PR8M, MOI = 1), and itaconate and DI treatments were given at the indicated concentrations. Expression of *HA* and *CXCL10* mRNA was measured by RT-qPCR 24 h p.i. (n = 3). Reference for fold change = uninfected cells 24 h. DI concentrations $\geq$ 1 mM could not be evaluated due to widespread cytotoxicity seen by light microscopy. Toxicity data of the compounds on A549 cells are shown in **S2 Fig**.
(TIF)

**S10 Fig. Global effects of itaconate and DI on transcriptomic responses in IAV-infected A549 cells.** A549 cells were infected with IAV (PR8M, MOI = 1) and gene expression was assessed by microarray analysis 24 h p.i. PCA based on the microarray analysis used for **S11**–**S13 Figs**, but additionally including treatment with 0.25 mM DI. The PCA shows normalization of IAV-driven reprogramming of gene expression, but increasing impact of itaconate on cellular responses with increasing doses. Toxicity data and dose-response curves of the compounds for HA mRNA and *CXCL10* mRNA expression are shown in **S2** and **S9 Figs**, respectively.
(TIF)

**S11 Fig. Effects of itaconate (20 mM) and DI (0.5 mM) on transcriptomic responses in IAV-infected A549 cells.** A549 cells were infected with IAV (PR8M, MOI = 1) in the presence or absence of itaconate or DI treatment, and gene expression was assessed by microarray analysis 24 h p.i. Analysis based on the microarray data used for the GO enrichment analysis in **S13 Fig**. Unsupervised hierarchical clustering analysis of the 100 most significant DEGs (FDR F-test <1.46E-10). DI-treated infected and uninfected cells cluster in one clade, and infected and itaconate-treated infected cells in the other. There is a large clade mostly containing pro-inflammatory genes, which are downregulated by DI, whereas effects of itaconate are much weaker. *DUSP95* is exclusively downregulated by itaconate (arrowhead). There are two smaller clades of genes that form an "itaconate signature", comprising genes that are, albeit to a lesser extent, also upregulated by IAV infection, or uniquely by itaconate.
(TIF)

**S12 Fig. Effects of itaconate (40 mM) and DI (0.5 mM) on transcriptomic responses in IAV-infected A549 cells.** A549 cells were infected with IAV (PR8M, MOI = 1) in the presence or absence of itaconate or DI treatment, and gene expression was assessed by microarray analysis 24 h p.i. Hierarchical clustering analysis, using a subset of the microarray analysis used for the PCA in **S10 Fig**. Compared to the 20 mM itaconate concentration (**S11 Fig**), itaconate-treated IAV infection is now in a separate clade, indicating that the impact of itaconate on the cells dominates that of the infection. Downregulation of the inflammation-driven clade is now more pronounced, the itaconate-unique signature is stronger, and there also is a clade comprised of genes downregulated by IAV.
(TIF)

**S13 Fig. GO term enrichment analysis of itaconate and DI reveals global effects on IFN-signaling and cellular inflammation in A549 cells.** Analysis performed based on the microarray data also used for the hierarchical clustering analysis in **S11 Fig.** Cells were, in the case of treatment, incubated overnight with DI (0.5 mM) or itaconate (20 mM), and then infected with IAV (PR8M, MOI = 1). Gene expression was determined by microarray analysis 24 h p.i.

(n = 4). **A**. GO enrichment (FDR <0.05) analysis using DEGs (p<0.05, FC >|1.5|) as input. GO terms are listed in reverse alphabetical order from top to bottom, with upright (red) triangles indicating enrichment (up-regulation) and downward pointing (blue) triangles depletion (down-regulation). **B**. Hierarchical cluster analysis illustrating downregulation of a subset of genes involved in IFNA/B signaling.
(TIF)

**S14 Fig. KEGG pathway enrichment analysis of itaconate (20 mM) and DI (0.5 mM) treatment on IAV infection in A549 cells.** Analysis performed based on the microarray data also used for the hierarchical clustering analysis in **S11 Fig**. A549 cells were infected with IAV (PR8M, MOI = 1) in the presence or absence of itaconate or DI treatment, and gene expression was assessed by microarray analysis 24 h p.i.
(TIF)

**S15 Fig. Primary human lung tissue explants support IAV RNA transcription and upregulation of *CXCL10* mRNA.** Human primary lung tissue from patients with emphysema or pulmonary arterial hypertension (n = 7 donors, 3 tissue pieces per donor per treatment) was infected with IAV (PR8M, MOI = 1) for 72 h and *HA* and *CXCL10* mRNA expression measured by RT-qPCR, using *HPRT1* as internal reference. Reference for fold change = uninfected tissue at the same time point. *p<0.05; **p<0.01; ***p<0.001 (Mann-Whitney U test).
(TIF)

**S16 Fig. Induction of *ACOD1* and *TNFAIP3* mRNAs during IAV (H3N2) infection of human lung explants.** Reanalysis of a published dataset of gene expression (RNAseq) in IAV A/Panama/2007/1999 (H3N2) infection (dose not specified) of human lung tissue derived from tumor-free margins obtained during lobectomy for lung carcinoma (39). A strong induction of *ACOD1* expression and a tendency towards increased *TNFAIP3* expression are seen. As opposed to the strong induction of both genes in the mouse model (**Fig 1A**), *HMOX1* and *HMOX2* expression is unchanged. *p<0.05; **p<0.01; ***p<0.001 (pairwise t-tests with pooled SD).
(TIF)

**S17 Fig. Increased levels of *ACOD1* and *TNFAIP3* mRNA expression in whole blood from patients with moderate and severe influenza.** Reanalysis of a published data set of gene expression in whole blood from patients with moderate and severe influenza and healthy controls [64]. *p<0.05; **p<0.01; ***p<0.001 (pairwise t-tests with pooled SD).
(TIF)

**S18 Fig. Transcriptome changes in PBMC treated with itaconate, DI, or 4OI are driven more by the treatments than by infection.** Analysis of mRNA expression in a subgroup (n = 4 donors) of the PBMC samples that were used for the targeted assays shown in **Fig 8**. PBMC were infected with IAV (PR8M, MOI = 1) and gene expression was analyzed by microarray analysis 12 h p.i. A PCA was performed on the same microarray data as used for the GO enrichment analysis in **Fig 8C**. There is a less pronounced effect of IAV infection (control and IAV infected samples are not clearly separated) than on dTHP-1 and A549 cells, but pronounced additional broad changes in gene expression due to treatment with 4OI, itaconate (note two outliers), and DI (note one outlier) are seen.
(TIF)

**S19 Fig. Hierarchical clustering analysis of transcriptomic changes in IAV infected PBMC and responses to itaconate, DI, and 4OI treatment.** PBMC were infected with IAV (PR8M, MOI = 1) and gene expression was analyzed by microarray analysis 12 h p.i. Analysis based on

the same microarray data as **Figs 8C** and **S18**. The 100 most significant DEGs were selected (FDR F-test <3.72E-05). Transcriptome changes are mostly due to marked effects of 4OI (more down- than up-regulation) on genes that are not affected by IAV infection, indicating general effects on cell homeostasis. However, induction of *HMOX1* by DI in IAV infection is evident (red arrowhead). The blue arrowhead points to the extreme outlier under DI treatment seen in the PCA (**S18 Fig**).
(TIF)

**S20 Fig. KEGG pathway analysis of transcriptomic changes in IAV infected PBMCs and responses to itaconate, DI, and 4OI treatment.** PBMCs were infected with IAV (PR8M, MOI = 1) and gene expression was analyzed by microarray analysis 12 h p.i. Analysis based on the same microarray data as **S18** and **S19 Figs**. KEGG terms with an FDR <0.05 in at least one group were selected. The threshold for inclusion in the chart was raised to FDR <0.01 if a pathway was enriched/depleted only in a single treatment and not by IAV infection alone (e.g., *Alcoholism* in 4OI treatment).
(TIF)

**S21 Fig. Effects of IAV infection and DI treatment on relative cell type distribution in PBMCs.** PBMCs were infected with IAV (PR8M, MOI = 1) and gene expression was assessed by scRNAseq 12 h p.i. Analysis based on the data set derived from the experiment shown in **Fig 9**. A reduction of CD14+ monocytes is apparent under DI treatment of both uninfected and infected PBMC. **A.** Percentages. **B.** Absolute numbers.
(TIF)

**S22 Fig. Effects of IAV infection and DI treatment on transcriptomes in the PBMC cell types not shown in Fig 9.** PBMCs were infected with IAV (PR8M, MOI = 1) and gene expression was assessed by scRNAseq 12 h p.i. Analysis based on the data set derived from the experiment shown in **Fig 9**. Volcano plots comparing effects of DI treatment, with and without IAV infection, on CD4+ cells, NK cells, and B cells.
(TIF)

**S23 Fig. Cell-type specific KEGG pathway analysis of effects of DI on infected and uninfected monocytes, CD4 T cells, CD8 T cells, B cells, and NK cells.** PBMCs were infected with IAV (PR8M, MOI = 1) and gene expression was assessed by scRNAseq 12 h p.i. Analyses based on the scRNAseq data used for **Figs 9** and **10**. Pathways that are enriched (FDR <0.05) in at least one cell type are shown.
(TIF)

**S24 Fig. Identification of central pathways commonly regulated in monocytes, CD4 and CD8 T cells, B cells, and NK cells in response to IAV infection and DI treatment.** PBMCs were infected with IAV (PR8M, MOI = 1) and gene expression was assessed by scRNAseq 12 h p.i. GO Biological Process (A) and KEGG (B) functional enrichment analyses of all genes commonly differentially expressed in monocytes, CD4 and CD8 T cells, B cells, and NK cells in response to IAV infection and DI treatment (i.e. the central intersect of the Venn diagram in **Fig 9G**).
(TIF)

**S25 Fig. DI treatment prevents histophathological alterations in lungs of IAV-infected mice.** Representative histological findings in lungs of IAV infected mice with or with DI treatment (50 mg/Kg once daily, IP) compared with mock-infected (PBS control) mice. Representative photomicrographs (H&E stains) of sections used for the inflammation score analysis shown in **Fig 11F**. **A,B**. IAV infection without treatment. **A.** Massive inflammatory infiltration

with neutrophil predominance, which fills the alveolar air spaces. **B.** Necrotizing bronchiolitis. There is necrosis of the bronchiolar wall, with submucosal edema and vascular congestion. The epithelial layer is desquamating, and necrotic epithelial cells are present in the lumen. Interstitial fibrosis with thickening of the muscular artery wall is present throughout. **C,D**. IAV infection with DI treatment. **C.** Normal appearing alveoli with preserved air spaces and no inflammatory infiltrates. **D.** Bronchiole without inflammation, but thickening of the muscular artery wall. **E,F**. Mock infection (PBS control). **E.** Normal appearing alveoli with preserved air spaces and no inflammatory infiltrates. **F.** Normal appearing bronchiole without signs of inflammation or wall thickening.
(TIF)

## Acknowledgments

We thank V. Kaever, A. Garbe, and staff of the Research Core Unit Metabolomics at Hannover Medical School for support with the itaconate measurements, Amandine Bernard (University of Luxembourg) for mouse genotyping, Elena Reinhard (HZI) for assistance with mouse experiments, Djalil Coowar (University of Luxembourg) and staff of the HZI Animal Facility for animal care and logistics, the HZI Genome Analytics Platform for microarray and RNA sequencing, L. Gröbe (HZI) for cell sorting, and D. Jonigk and P. Braubach (Institute of Pathology, Hannover Medical School) for providing human lung tissue. Furthermore, we thank Hans Jörg Hauser (HZI) for helpful comments on the first draft of the manuscript. We are particularly grateful to Prof. Abdelrazik (NRC Cairo, Egypt) for contributing the histological data shown in **Figs 11F** and **S25**.

## Author Contributions

**Conceptualization:** Azeem A. Iqbal, Nishika Sahini, Mohamed Tantawy, Ahmed Mostafa, Stephan Pleschka, Alessandro Michelucci, Frank Pessler.

**Data curation:** Nishika Sahini, Moritz Winterhoff, Thomas Ebensen, Robert Geffers.

**Formal analysis:** Aaqib Sohail, Azeem A. Iqbal, Nishika Sahini, Mohamed Tantawy, Syed F.H. Waqas, Moritz Winterhoff, Thomas Ebensen, Kristin Schultz, Robert Geffers, Klaus Schughart, Marina C. Pils, Christine Falk, Frank Pessler.

**Funding acquisition:** Stephan Pleschka, Frank Pessler.

**Investigation:** Aaqib Sohail, Azeem A. Iqbal, Nishika Sahini, Fangfang Chen, Mohamed Tantawy, Syed F.H. Waqas, Thomas Ebensen, Kristin Schultz, Matthias Preusse, Mahmoud Shehata, Marina C. Pils, Ahmed Mostafa.

**Methodology:** Azeem A. Iqbal, Nishika Sahini, Fangfang Chen, Mohamed Tantawy, Syed F.H. Waqas, Moritz Winterhoff, Thomas Ebensen, Kristin Schultz, Robert Geffers, Matthias Preusse, Mahmoud Shehata, Heike Bähre, Marina C. Pils, Ahmed Mostafa, Stephan Pleschka, Frank Pessler.

**Project administration:** Frank Pessler.

**Resources:** Heike Bähre, Carlos A. Guzman, Ahmed Mostafa, Stephan Pleschka, Christine Falk, Alessandro Michelucci, Frank Pessler.

**Supervision:** Carlos A. Guzman, Frank Pessler.

**Validation:** Klaus Schughart.

**Visualization:** Azeem A. Iqbal, Nishika Sahini, Fangfang Chen, Syed F.H. Waqas, Moritz Winterhoff, Klaus Schughart, Matthias Preusse.

**Writing – original draft:** Azeem A. Iqbal, Nishika Sahini, Mohamed Tantawy, Thomas Ebensen, Robert Geffers, Frank Pessler.

**Writing – review & editing:** Aaqib Sohail, Fangfang Chen, Mohamed Tantawy, Syed F.H. Waqas, Moritz Winterhoff, Thomas Ebensen, Kristin Schultz, Matthias Preusse, Mahmoud Shehata, Marina C. Pils, Carlos A. Guzman, Ahmed Mostafa, Stephan Pleschka, Frank Pessler.

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
