## [Decision Letter · Decision Letter 0]

22 Mar 2021

Dear Dr. Pessler,

Thank you very much for submitting your manuscript "Itaconate and derivatives reduce interferon responses and inflammation in influenza A virus infection" for consideration at PLOS Pathogens. As with all papers reviewed by the journal, your manuscript was reviewed by members of the editorial board and by several independent reviewers. In light of the reviews (below this email), we would like to invite the resubmission of a significantly-revised version that takes into account the reviewers' comments.

A major criticism is that the data provided is mainly descriptive, and some key additional experiments are required, i.e. in additional mouse experiments where inflammatory responses and viral titers are measured (see reviewer 1 comments). All three reviewers raised concerns that the data was not well-presented and the manuscript requires major re-structuring for clarity. A summary schematic would also be helpful.

We cannot make any decision about publication until we have seen the revised manuscript and your response to the reviewers' comments. Your revised manuscript is also likely to be sent to reviewers for further evaluation.

Sincerely,

Meike Dittmann, Ph.D.

Associate Editor

PLOS Pathogens

Sonja Best

Section Editor

PLOS Pathogens

Kasturi Haldar

Editor-in-Chief

PLOS Pathogens

orcid.org/0000-0001-5065-158X

Michael Malim

Editor-in-Chief

PLOS Pathogens

orcid.org/0000-0002-7699-2064

A major criticism is that the data provided is mainly descriptive, and some key additional experiments are required, i.e. in additional mouse experiments where inflammatory responses and viral titers are measured (see reviewer 1 comments). All three reviewers raised concerns that the data was not well-presented and requires re-structuring for clarity. A summary schematic would also be helpful.

Reviewer's Responses to Questions

**Part I - Summary**

Reviewer #1: The manuscript by Sohail and colleagues investigated the inflammatory response of itaconate and itaconate derivatives upon infection of influenza A virus. The authors treated itaconate and derivatives to cells infected with the virus and observed that interferon response was repressed without affecting viral replication. Further, they compared gene expression in treated cells and identified gene signatures induced by each compound (i.e. itaconate, DMI, and 4OI). The findings may provide useful information on the complex inflammatory response upon virus infection and how itaconate and its derivative influence inflammation linking itaconate molecules to inflammatory response and disease outcome.

The impact of itaconate and chemical derivatives on the immune response is quite interesting. However, the specific mechanism at play and the distinct effect of itaconate and chemical derivatives are not clear. Recent studies have demonstrated that itaconate and itaconate derivatives have a distinct impact on metabolism, signaling (i.e. Nrf2), and immune response, specifically recent work by the Artyomov lab (PMID: 32694786). As such, itaconate and itaconate derivatives should not be used interchangeably and might be better discussed as different compounds. Some additional studies would help to clarify these issues, as the overall impact on inflammation is strong.

Reviewer #2: Sohail and colleagues describe in this manuscript the effects of itaconate (Ita) and its derivatives dimethyl-itaconate (DI) and 4-octyl-itaconate (4OI). Ita is a metabolite made by aconitate decarboxylase, which is encoded by the gene ACOD1. Ita is synthesized from cis-aconitate, a metabolite in the TCA cycle. Ita’s role in modulating inflammatory responses that occur in microbial infections is well document. Additionally, Ita and some of its derivatives have also been studied with regards to viral infection, including influenza. In the current manuscript, the authors explore the possible role of Ita, DI, and 4OI in regulating host responses to infection, including pro- and anti-inflammatory responses. The work has a lofty goal of reducing the harmful damage caused by the host immune response to flu infection while not creating a favorable environment for the virus. While the submitted work does not reach that goal, it provides a necessary step to testing if Ita, DI, or 4OI has the potential to reduce inflammation while not favoring virus replication. The authors use ACOD1 knockout mice, human PBMCs, macrophages derived from the monocytic THP-1 cell line, A549 cells, and human lung tissue explant models. The work relies heavily on measurements from gene expression, including RNAseq and single-cell RNAseq. The involves using ELISA and cytokine panels to measure the effects of Ita, DI, and 4OI treatment on uninfected and IAV infected cells. The experiments appear to be well controlled, including adjusting the pH of the medium after the addition of IA, DI, or 4OI, which is necessary to draw appropriate conclusions from this type of work. Given the number of infection models / cell types examined in the manuscript, it was helpful that the authors included this information in the titles of the figure legends. Since Ita and its derivates are likely to be important to the host response to infection with other viruses and some bacteria, the study will likely be of broad interest to other microbiologists and immunologists. This study is a comprehensive, mostly observational study. While the work is highly-descriptive, it is likely to lead to more mechanistic findings in the future. I have no significant concerns with work as a whole. At times, the data is presented in a way that distracts from the work, so I provide comments focused on improving the clarity of the work.

Reviewer #3: This study by Sohail et al is centered upon investigating the role of itoconate pathway in regulating influenza virus induced inflammatory pathway. They use either Ita or DI treatment in several different viral infected cell lines, human lung culture system, PBMCs and perform RNA seq analysis and show that inflammatory gene expression is reduced in Ita or DI treated infected cells when compared to vehicle treated infected cells. In particular CXCL10 and MCP1 (CCL2) gene transcript are reduced in most cell types tested. Acod1-/- mice are also used to show that Acod1 deficiency is protective during influenza virus infection; the mice lose less weight and survive better when compared to WT mice.

Major concerns:

The study as presented is overly descriptive and utilizes several different cell lines in vitro to perform transcriptomic analysis on each cell line following treatment with Ita and DI. There is no attempt made to understand even at a very superficial level as to what the physiological consequences are of treatment with Ita or DI. CXL10 and CCL2 are potent chemoattractants of monocytes and other myeloid as well as T cells. The authors should attempt to at the very least perform these studies in vivo in the mouse model and use flow cytometry to determine how the inflammatory response is affected by these treatments and if downregulation of CXCL10 and CCL2 transcripts matches actual reduction in protein production and a concomitant decrease in myeloid cell (monocytes, neutrophils etc) recruitment. In addition, macrophages are important players in regulating and mediating the inflammatory process after influenza infection in the lungs. As demonstrated by the investigators Acod1 gene expression is particularly high in M2 macrophages. There are many subsets of macrophages in the lungs that mediate distinct functions, the investigators should perform a more sophisticated analysis of macrophage subset activation status in the lungs of mice infected with influenza using WT and Acod1-/- mice. Furthermore, no analysis of viral titers is performed in Acod1-/- mice. Is the increased death and morbidity observed in Acod1-/- mice due to increase in viral replication? This should be directly demonstrated. Analysis of HA transcripts in a cell line is not sufficient.

An analysis of the cytokines in the BAL and lungs using multiplex (luminex) approach would also be beneficial to understand the mechanisms that are important for the observed protection in Acod1 deficient mice.

The study suffers from repeated genomic studies performed on several different cell lines with similar results. Pretty much all that data can be moved to the supplementary section. The exact reasons for using so many different cells lines is not clear and does not add significance to the study. The paper is overtly unfocused and repetitious and hard to understand what the exact conclusions are.

Figs 2 and 5 have several parts missing, thus, it is hard to evaluate this study when two main figures have several parts missing.

Minor concerns:

The paper will benefit if the authors describe some of the studies better in the results section. It is difficult to assess exactly what was and done in each experiment. For example how were the M1 vs M2 macrophages generated?

FigS3- Please annotate the heatmaps for that entire cluster on the bottom. It has some important genes like TGFb and S100a8, the importance of these genes is not discussed.

FigS4- Can the conditions be put together. heat maps throughout this paper have confusing color scheme, can they change it to make it black and white for one or the other?

Fig-4: This figure is too busy, needs to be simplified with a better form. Difficult to follow along.

**Part II – Major Issues: Key Experiments Required for Acceptance**

Reviewer #1: 1. The authors stated that “chemical variants of itaconate, rather than its native form, will take the lead in further translational development to the bedside” (line 428). The “cytoprotective properties” and the mechanism of action of itaconate and chemical derivatives are not well understood. It became clear that itaconate and itaconate derivatives have very different impacts on metabolism (specifically SDH inhibition, see Artyomov lab PMID: 32694786 ). As such, the primary mechanism of action is not well understood. At times, the authors may reframe their statements; for instance, “both compounds likely exert cytoprotective effects by reducing ROS generation via inhibition of SDH.” (line 102).

2. Interestingly, D2 mice models have decreased ACOD1 expression compared to B6 mice. Does reduced ACOD1 expression led to reduced itaconate levels in the D2 animal model? Expression level does not reflect metabolite level, and itaconate concentrations might be similar in both animal models. If itaconate levels are identical, ACOD1/itaconate might not be the primary driver for the different disease severity described in the two animal models. Further, some quantification of intracellular itaconate levels would be beneficial to better understand the treatment doses chosen for this study as 25 mM or 40 mM extracellular itaconate seems to be very high. Are those concentrations physiological relevant? Specifically, which itaconate concentrations are achieved in macrophages and PBMCs upon infection (Fig. 2)?

3. Itaconate derivatives seem to decrease ACOD1 expression levels (Fig 8). Some effects observed by treatments with itaconate derivatives might be due to decreased endogenous itaconate synthesis. On the other hand, 4OI might be converted into itaconate (Hooftman et al. 2020, study with 13C 4OI). How do itaconate levels change in response to DMI and 4OI treatments? Further, the authors may want to repeat some key experiments in IRG1KO cells treated with DMI or 4OI to decipher the impact of itaconate derivatives compared to endogenous itaconate levels.

4. The authors pretreated cells for 24h before infection (Fig 3) for most of their studies. Are similar results achieved with acute treatments (as opposed to pretreatments)? Pretreatments might be clinically challenging.

Reviewer #2: 1. At times, the GO/KEGG enrichment analyses provide more distraction than clarity. It is unclear why some GO terms were included in the main figures, while others were placed in the supplement. For example, why include “excretion” in Fig. 4 while putting “Influenza A”, “RIG-I-like signaling”, and “TNF signaling” in Fig. S5? As shown in Fig. S10 and S16, why would IAV infection show a greater enrichment for “Herpes simplex infection” and “Hepatitis C” than “Influenza A”? The authors may want to re-examine if including all of the GO/KEGG analyses is necessary.

2. This is a more of a comment than a major criticism. The authors expend a great deal of effort comparing and contrasting Ita treatment to DI or 4OI treatment. Since this work is descriptive work, it provides little insight into potential mechanistic differences between their modes of action. However, the authors’ data reveal interesting observations regarding the potential effects of DI and 4OI treatment on ACOD1 expression (and possibly intracellular Ita concentrations) that is largely ignored. Fig. 8 provides the best observation in this regard. In Fig. 8, the authors show that IAV infection induces the expression of ACOD1 in PBMCs (a finding that confirms observations in Fig. 2). Ita treatment further promotes ACOD1 expression in infected cells. However, DI and 4OI reduce ACOD1 expression in infected cells and uninfected cells. A similar observation is found in Fig. 9, where infection increases ACOD1 expression in monocytes, but DI treatment reduces it. Did the authors measure intracellular Ita concentrations in these conditions to see if they correlate with ACOD1 expression? It would be interesting to know if the authors think that DI and 4OI affect on ACOD1 expression (and presumably intracellular Ita concentrations) is a possible mechanism that explains the differences they observed between Ita treatment and DI/4OI treatment.

Reviewer #3: As noted above

The authors should attempt to at the very least perform these studies in vivo in the mouse model and use flow cytometry to determine how the inflammatory response is affected by these treatments and if downregulation of CXCL10 and CCL2 transcripts matches actual reduction in protein production and a concomitant decrease in myeloid cell (monocytes, neutrophils etc) recruitment.

In addition, macrophages are important players in regulating and mediating the inflammatory process after influenza infection in the lungs. As demonstrated by the investigators Acod1 gene expression is particularly high in M2 macrophages.

There are many subsets of macrophages in the lungs that mediate distinct functions, the investigators should perform a more sophisticated analysis of macrophage subset activation status in the lungs of mice infected with influenza using WT and Acod1-/- mice. Furthermore, no analysis of viral titers is performed in Acod1-/- mice. Is the increased death and morbidity observed in Acod1-/- mice due to increase in viral replication? This should be directly demonstrated. Analysis of HA transcripts in a cell line is not sufficient.

An analysis of the cytokines in the BAL and lungs using multiplex (luminex) approach would also be beneficial to understand the mechanisms that are important for the observed protection in Acod1 deficient mice.

**Part III – Minor Issues: Editorial and Data Presentation Modifications**

Reviewer #1: The authors may cite and discuss some key studies on itaconate derivatives, specifically work by Artyomov Lab (PMID: 32694786, impact on inflammation), Hooftman et a (PMID: 32791101, Cell Metab, inhibition of NLRP2 inflammasome activation by 4OI), and Sethy et al. (2019, J.Med.Chem, identification of itaconate derivates as potential anti-influenza agents). For instance, the current manuscript demonstrated that CXCL10 expression was decreased upon itaconate/derivative treatment, while work by Artyomov lab reported increased levels upon itaconate treatment in IRG1KO conditions.

Reviewer #2: 1. The manuscript would benefit from a model or schematic to summarize the most salient data clarify their overall conclusion(s).

2. In Fig. 7A, Ita and DI treatments were found to have no effect on virus titers in the human lung tissue explant model. This observation is mostly ignored in the rest of the manuscript. This is a significant finding; the authors should include this information in the Abstract and Discussion.

3. The authors discuss the cytotoxicity of Ita, DI, and OA4 treatment on several cell types on several occasions, but no data were presented. The authors need to include the data from the cytotoxicity assays in the manuscript.

4. At the end of the description for Fig. 8 the authors state that “the miscellaneous effects of 4OI on gene expression in this model, we subsequently focused on DI to decipher effects at the single-cell level.” However, 4OI had what appears to be a stronger anti-inflammatory property than DI, while having the added benefit of reducing HA expression. Could the author expand on what exactly they mean by “miscellaneous effects of 4OI on gene expression” that caused them to focus on DI instead of 4OI?

5. Line 299: “DI treatment markedly reduced expression of ACOD1, IFIT1, and CXCL10, but –interestingly– it reduced TNFAIP3 and ISG15 (albeit weakly) expression only in monocytes.” But DI treatment caused a loss of monocytes. In the above state are the authors implying that there is a reduction in TNFAIP3, IFIT1, ACOD1, and ISG15 (weakly) in the overall levels because of the loss in monocyte numbers, or is there a reduction in their expression on a per monocyte cell basis?

6. The authors should include a discussion/reference to Sethy, et al., “Design, Synthesis, and Biological Evaluation of Itaconic Acid Derivatives as Potential Anti-Influenza Agents” https://doi.org/10.1021/acs.jmedchem.8b01683

7. Could the authors clarify why they used only female mice?

8. Lines 242-244 “infection with IAV led to brisk transcription of viral HA and host CXCL10 mRNA in a 72 h time course (Fig. S11), whereas ACOD1 mRNA levels did not change significantly.” No ACOD1 data is presented in Fig. S11, and this statement appears to be in contradiction to data shown in Fig. S12.

9. The figure legend for Fig. 1G is missing details. Are the images shown from WT or KO mice? Also, there is no reference to “Fig. 1G” in the text of the result section.

10. Throughout the manuscript “FC” is used in the figures. But it is often unclear exactly how the fold change was calculated. A short description in the figure legend each time “FC” appears would help.

11. The figure legend for Fig. 3 is missing a description of the concentrations of DI used in the experiment.

12. A description of the generation of the A549 cells expressing ACOD1 appears to be missing from the Materials and Methods section.

13. Fig. 2 only has a panel A and B. However, the text describing Fig. 2 references Figure C-E. This makes the description on lines 151-163 somewhat unclear.

14. The authors need to describe Fig. 3A in the text of the Results section.

15. Line 212: “IA” do they mean itaconate? In other places, they abbreviate it “Ita”.

16. Figure 5 only has a panel A and B. However, the text describing Figure 5 references Figure C-H. This makes the description on lines 206-223 somewhat unclear.

17. It would help if additional details regarding the infection were included in the figure legends that included M.O.I. and time of infection, particularly when performing reanalysis of published data (for example, Fig. 1A).

18. Line 341. “ell” typo

19. Line 763: “Go” typo

20. Reference 50 needs greater detail.

Reviewer #3: The paper will benefit if the authors describe some of the studies better in the results section. It is difficult to assess exactly what was and done in each experiment. For example how were the M1 vs M2 macrophages generated?

FigS3- Please annotate the heatmaps for that entire cluster on the bottom. It has some important genes like TGFb and S100a8, the importance of these genes is not discussed.

FigS4- Can the conditions be put together. heat maps throughout this paper have confusing color scheme, can they change it to make it black and white for one or the other?

Fig-4: This figure is too busy, needs to be simplified with a better form. Difficult to follow along.

PLOS authors have the option to publish the peer review history of their article (what does this mean?). If published, this will include your full peer review and any attached files.

Reviewer #1: No

Reviewer #2: No

Reviewer #3: No
---

## [Editor Report · Decision Letter 1]

17 Dec 2021

Dear Dr. Pessler,

We are pleased to inform you that your manuscript 'Itaconate and derivatives reduce interferon responses and inflammation in influenza A virus infection' has been provisionally accepted for publication in PLOS Pathogens.

Best regards,

Meike Dittmann, Ph.D.

Associate Editor

PLOS Pathogens

Sonja Best

Section Editor

PLOS Pathogens

Kasturi Haldar

Editor-in-Chief

PLOS Pathogens

orcid.org/0000-0001-5065-158X

Michael Malim

Editor-in-Chief

PLOS Pathogens

orcid.org/0000-0002-7699-2064
---

## [Editor Report · Acceptance letter]

7 Jan 2022

Dear Dr. Pessler,

We are delighted to inform you that your manuscript, "Itaconate and derivatives reduce interferon responses and inflammation in influenza A virus infection," has been formally accepted for publication in PLOS Pathogens.

Best regards,

Kasturi Haldar

Editor-in-Chief

PLOS Pathogens

orcid.org/0000-0001-5065-158X

Michael Malim

Editor-in-Chief

PLOS Pathogens

orcid.org/0000-0002-7699-2064